# Vector-valued Distance and Gyrocalculus on the Space of Symmetric Positive Definite Matrices

**Federico López**[*]
HITS

**Beatrice Pozzetti**
Heidelberg University

**Steve Trettel**
Stanford University

**Michael Strube**
HITS

**Anna Wienhard**
Heidelberg University
HITS

## Abstract

We propose the use of the vector-valued distance to compute distances and extract geometric information from the manifold of symmetric positive definite matrices (SPD), and develop gyrovector calculus, constructing analogs of vector space operations in this curved space. We implement these operations and showcase their versatility in the tasks of knowledge graph completion, item recommendation, and question answering. In experiments, the SPD models outperform their equivalents in Euclidean and hyperbolic space. The vector-valued distance allows us to visualize embeddings, showing that the models learn to disentangle representations of positive samples from negative ones.

## 1 Introduction

Symmetric Positive Definite (SPD) matrices have been applied in many tasks in computer vision such as pedestrian detection [75, 79], action [36, 51, 60] or face recognition [41, 42], object [43, 90] and image set classification [86], visual tracking [87], and medical imaging analysis [5, 65] among others. They have been used to capture statistical notions (Gaussian distributions [68], covariance [78]), while respecting the Riemannian geometry of the underlying SPD manifold, which offers a convenient trade-off between structural richness and computational tractability [27]. Previous work has applied approximation methods that locally flatten the manifold by projecting it to its tangent space [22, 83], or by embedding the manifold into higher dimensional Hilbert spaces [35, 90].

These methods face problems such as distortion of the geometrical structure of the manifold and other known concerns with regard to high-dimensional spaces [29]. To overcome these issues, several distances on SPD manifolds have been proposed, such as the Affine Invariant metric [65], the Stein metric [71], the Bures–Wasserstein metric [13] or the Log-Euclidean metric [5, 6], with their respective geometric properties. However, the representational power of SPD is not fully exploited in many cases [6, 65]. At the same time, it is hard to translate operations into their non-Euclidean domain given the lack of closed-form expressions. There has been a growing need to generalize basic operations, such as addition, rotation, reflection or scalar multiplication, to their Riemannian geometric counterparts to leverage this structure in the context of Geometric Deep Learning [19].

SPD manifolds have a rich geometry that contains both Euclidean and hyperbolic subspaces. Thus, embeddings into SPD manifolds are beneficial, since they can accommodate hierarchical structure in data sets in the hyperbolic subspaces while at the same time represent Euclidean features. This makes them more versatile than using only hyperbolic or Euclidean spaces, and in fact, their different submanifold geometries can be used to identify and disentangle such substructures in graphs.

---

[*]Correspondence to `federico.lopez@h-its.org`

In this work, we introduce the vector-valued distance function to exploit the full representation power of SPD (§3.1). While in Euclidean or hyperbolic space the relative position between two points is completely captured by their distance (and this is the only invariant), in SPD this invariant is a vector, encoding much more information than a single scalar. This vector reflects the higher expressivity of SPD due to its richer geometry encompassing Euclidean as well as hyperbolic spaces. We develop algorithms using the vector-valued distance and showcase two main advantages: its versatility to implement universal models, and its use in explaining and visualizing what the model has learned.

Furthermore, we bridge the gap between Euclidean and SPD geometry by developing gyrocalculus in SPD (§4), which yields closed-form expressions of arithmetic operations, such as addition, scalar multiplication and matrix scaling. This provides means to translate previously implemented ideas in different metric spaces to their analog notions in SPD. These arithmetic operations are also useful to adapt neural network architectures to SPD manifolds.

We showcase this on knowledge graph completion, item recommendation, and question answering. In the experiments, the proposed SPD models outperform their equivalents in Euclidean and hyperbolic space (§6). These results reflect the superior expressivity of SPD, and show the versatility of the approach and ease of integration with downstream tasks.

The vector-valued distance allows us to develop a new tool for the analysis of the structural properties of the learned representations. With this tool, we visualize high-dimensional SPD embeddings, providing better explainability on what the models learn (§6.4). We show that the knowledge graph models are capable of disentangling and clustering positive triples from negative ones.

## 2   Related Work

Symmetric positive definite matrices are not new in the Machine Learning literature. They have been used in a plethora of applications [5, 29, 36, 40–43, 51, 60, 65, 68, 75, 78, 79, 86, 87, 90]), although not always respecting the intrinsic structure or the positive definiteness constraint [22, 31, 35, 83, 90]. The alternative has been to map manifold points onto a tangent space and employ Euclidean-based tools. Unfortunately, this mapping distorts the metric structure in regions far from the origin of the tangent space affecting the performance [44, 96].

Previous work has proposed alternatives to the basic neural building blocks respecting the geometry of the space. For example, transformation layers [29, 33, 40], alternate convolutional layers based on SPDs [94] and Riemannian means [23], or appended after the convolution [21], recurrent models [24], projections onto Euclidean spaces [50, 57] and batch normalization [20]. Our work follows this line, providing explicit formulas for translating Euclidean arithmetic notions into SPDs.

Our general view, using the vector-valued distance function, allows us to treat Riemannian and Finsler metrics on SPD in a unified framework. Finsler metrics have previously been applied in compressed sensing [30], information geometry [69], for clustering categorical distributions [63], and in robotics [67]. With regard to optimization, matrix backpropagation techniques have been explored [2, 17, 43], with some of them accounting for different Riemannian geometries [20, 40]. Nonetheless, we opt for tangent space optimization [26] by exploiting the explicit formulations of the exponential and logarithmic map.

## 3   The Space $\mathrm{SPD}_n$

The space $\mathrm{SPD}_n$ is a Riemannian manifold of non-positive curvature of $n(n+1)/2$ dimensions. Points in $\mathrm{SPD}_n$ are positive definite real symmetric $n \times n$ matrices, with the identity matrix $I$ being a natural basepoint. The tangent space to any point of $\mathrm{SPD}_n$ can be identified with the vector space $S_n$ of all real symmetric $n \times n$ matrices. $\mathrm{SPD}_n$ contains $n$-dimensional Euclidean subspaces, $(n-1)$-dimensional hyperbolic subspaces as well as products of $\lfloor \frac{n}{2} \rfloor$ hyperbolic planes (see Helgason [39] for an in-depth introduction).

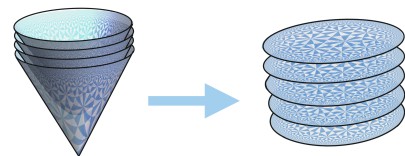

Figure 1: $\mathrm{SPD}_2$ is foliated by hyperboloids, each of which is a copy of the hyperbolic plane.

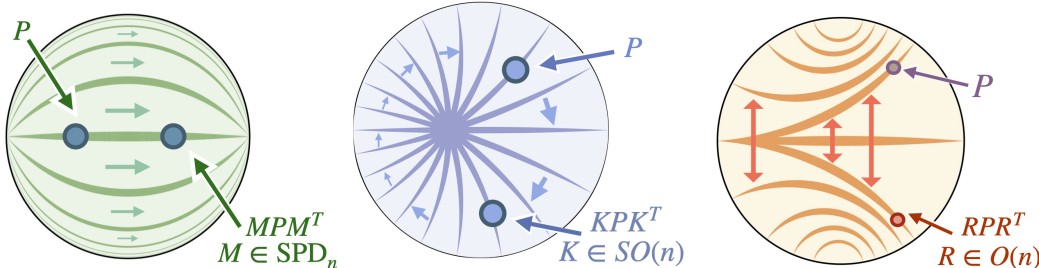

Figure 2: Some isometries of $\mathrm{SPD}_n$ have analogous Euclidean counterparts. Translation (left), rotation (center) and reflection (right).

In Figure 1 we visualize the smallest nontrivial example. $\mathrm{SPD}_2$ identifies with the inside of a cone in $\mathbb{R}^3$, cut out by requiring both eigenvalues of the matrix $\left(\begin{smallmatrix} x & y \\ y & z \end{smallmatrix}\right)$ to be positive. It carries the product geometry of the hyperbolic plane times a line.

**Exponential and logarithmic maps:** The exponential map, $\exp\colon S_n \to \mathrm{SPD}_n$, is a homeomorphism which links the Euclidean geometry of the tangent space $S_n$ and the curved geometry of $\mathrm{SPD}_n$. Its inverse is the logarithm map, $\log\colon \mathrm{SPD}_n \to S_n$. This pair of functions give diffeomorphisms that allows one to freely move between 'tangent space coordinates' or the original 'manifold coordinates'. We apply both maps based at $I \in \mathrm{SPD}_n$. The reason for this is that while mathematically any two points on $\mathrm{SPD}_n$ are equivalent, and we could obtain a different concrete expression for any other choice of basepoint $B \in \mathrm{SPD}_n$, the resulting formulas would be more complicated, and thus $I$ is the best choice from a computational point of view. We prove this in Appendix C.3.

**Symmetries:** The prototypical symmetries of $\mathrm{SPD}_n$ are parameterized by elements of $GL(n;\mathbb{R})$: any invertible matrix $M$ defines the symmetry $P \mapsto MPM^T$ acting on all points $P \in \mathrm{SPD}_n$. Thus many geometric transformations $\mathrm{SPD}_n$ can be completed using standard optimized matrix algorithms as opposed to custom-built procedures. See Appendix C.2. for a brief review of these symmetries.

Among these, we may find $\mathrm{SPD}_n$-generalizations of familiar symmetries of Euclidean geometry. When also $M$ is an element of $\mathrm{SPD}_n$, the symmetry $P \mapsto MPM^T$ is a generalization of an Euclidean *translation*, fixing no points of $\mathrm{SPD}_n$. When $M$ is an orthogonal matrix, the symmetry $P \mapsto MPM^T$ is conjugation by $M$, and thus fixes the basepoint $I = MIM^T = MM^{-1} = I$. We think of elements fixing the basepoint as being $\mathrm{SPD}_n$-*rotations* or $\mathrm{SPD}_n$-*reflections*, when the matrix $M$ is a familiar rotation or reflection (see Figure 2).

The Euclidean symmetry of *reflecting in a point* also has a natural generalization to $\mathrm{SPD}_n$. Euclidean reflection in the origin is given by $p \mapsto -p$; and its $\mathrm{SPD}_n$-analog, reflection in the basepoint $I$, is matrix inversion $P \mapsto P^{-1}$. The general $\mathrm{SPD}_n$-reflection in a point $Q \in \mathrm{SPD}_n$ is a conjugate of this by an $\mathrm{SPD}_n$ translation, given by $P \mapsto QP^{-1}Q$.

## 3.1 Vector-valued Distance Function

In Euclidean or hyperbolic spaces, the relative position between two points is completely determined by their distance, which is given by a scalar. For the space $\mathrm{SPD}_n$, it is determined by a vector.

**The VVD vector:** To assign this vector in SPD we introduce the vector-valued distance (VVD) function $d_{vv}\colon \mathrm{SPD}_n \times \mathrm{SPD}_n \to \mathbb{R}^n$. For two points $P, Q \in \mathrm{SPD}_n$, the VVD is defined as:

$$d_{vv}(P,Q) = \log(\lambda_1(P^{-1}Q), \dots, \lambda_n(P^{-1}Q)) \tag{1}$$

where $\lambda_1(P^{-1}Q) \geq \dots \geq \lambda_n(P^{-1}Q)$ are the eigenvalues of $P^{-1}Q$ sorted in descending order. This vector is an invariant of the relative position of two points up to isometry. This means that in $\mathrm{SPD}_n$, only if the VVD between two points $A$ and $B$ is the vector $v \in \mathbb{R}^n$, and the VVD between $P$ and $Q$ is also $v$, then there exists an isometry mapping $A$ to $P$ and $B$ to $Q$. Thus, we can recover completely the relative position of two points in $\mathrm{SPD}_n$ from this vector. For example, the Riemannian metric is obtained by using the standard $l_2$ norm on the VVD vector. This is: $d^R(P,Q) = ||d_{vv}(P,Q)||_2$. See [46] §2.6 and Appendix C.4 for a review of VVDs in symmetric spaces.

**Finsler metrics:** Any norm on $\mathbb{R}^n$ that is invariant under permutation of the entries induces a metric on $\mathrm{SPD}_n$. Moreover, $\mathrm{SPD}_n$ do not only support a Riemannian metric, but also Finsler metrics, a whole family of distances with the same symmetry group (group of isometries). These metrics are of special importance since distance minimizing geodesics are not necessarily unique in Finsler geometry. Two

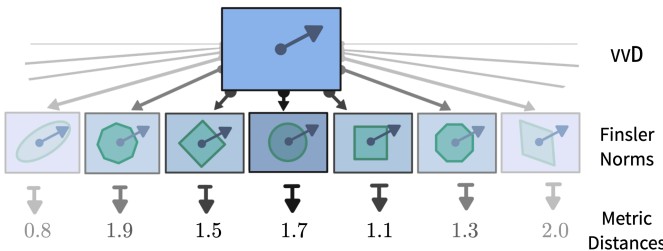

Figure 3: The vector-valued distance allows to reconstruct the Riemannian, or any Finsler distance.

different paths can have the same minimal length. This is particularly valuable when embedding graphs in $\mathrm{SPD}_n$, since in graphs there are generally several shortest paths. We obtain the Finsler metrics $\mathrm{F}_1$ or $\mathrm{F}_\infty$ by taking the respective $\ell_1$ or $\ell_\infty$ norms of the VVD in $\mathbb{R}^n$ (see Figure 3). See Planche [66] and Appendix C.6 for a review of the theory of Finsler metrics, [12] for the study of some Finsler metrics on $\mathrm{SPD}_n$, and [52] for applications of Finsler metrics on symmetric spaces in representation learning.

**Advantages of VVD:** The proposed metric learning framework based on the VVD offers several advantages. First, a single model can be run with different metrics, according to the chosen norm. The VVD contains the full information of the Riemannian distance and of all invariant Finsler distances, hence we can easily recover the Riemannian metric and extend the approach to many other alternatives (in Appendix C.7, we detail how the VVD generalizes other $\mathrm{SPD}_n$ metrics). Second, the VVD provides much more information than just the distance, and can be used to analyze the learned representation in $\mathrm{SPD}_n$, independent of the choice of the metric. Out of the VVD between two points, one can immediately read the regularity of the unique geodesics joining these two points. Geodesics in $\mathrm{SPD}_n$ have different regularity, which is related with the number of maximal Euclidean subspaces that contain the geodesic. The Riemannian or Finsler distances cannot distinguish the differences between these geodesics of different regularity, but the VVD function can. Third, the VVD function can be leveraged as a tool to visualize and analyze the learned high-dimensional representations (see §6.4).

## 4 Gyrocalculus

To build an analog of many Euclidean operators in $\mathrm{SPD}_n$, we require also a translation of operations internal to Euclidean geometry, chief among these being the vector space operations of addition and scalar multiplication. By means of tools introduced in pure mathematics literature[2], we describe a gyro-vector space structure on $\mathrm{SPD}_n$, which provides geometrically meaningful extensions of these vector space operations, extending successful applications of this framework in geometric deep learning to hyperbolic space [32, 54, 70]. These operations provide a template for translation, where one may attempt to replace $+, -, \times$ in formulas familiar from Euclidean spaces with the analogous operations $\oplus, \ominus, \otimes$ on $\mathrm{SPD}_n$. While straightforward, such translation requires some care, as gyro-addition is neither commutative nor associative. See Appendix D for a review of the underlying general theory of gyrogroups and additional guidelines for accurate formula translation.

**Addition and Subtraction:** Given a fixed choice $I$ of basepoint and two points $P, Q \in \mathrm{SPD}_n$, we define the gyroaddition of $P$ and $Q$ to be the point $P \oplus Q \in \mathrm{SPD}_n$ which is the image of $Q$ under the isometry which translates $I$ to $P$ along the geodesic connecting them. This directly generalizes the gyroaddition of hyperbolic space exploited by [7, 32, 70], via the geometric interpretation of Vermeer [84] (see Figure 4).

Fixing $P \in \mathrm{SPD}_n$, we may compute the value of $P \oplus Q$ for arbitrary $Q$ as the result of applying the $\mathrm{SPD}_n$-translation moving the basepoint to $P$, evaluated on $Q$. We see also the additive inverse of a point with respect to this operation must then be given by its geodesic reflection in $I$.

$$P \oplus Q = \sqrt{P}Q\sqrt{P} \qquad \ominus P = P^{-1} \qquad (2)$$

---

[2]See [1, 37] for an abstract treatment of gyrocalculus, and [47] for the specific examples discussed here.



Figure 4: Gyro-addition (left), gyro-scalar multiplication (center) and matrix scaling (right).

As this operation encodes a symmetry of $\mathrm{SPD}_n$, it is possible to recast certain geometric statements purely in the gyrovector formalism. In particular, the vector-valued distance $d_{vv}(P, Q)$ may be computed as the logarithm of the eigenvalues of $\ominus P \oplus Q$ (see Appendix C.5).

**Scalar Multiplication and Matrix Scaling:** For a fixed basepoint $I$, we define the scalar multiplication of a point $P \in \mathrm{SPD}_n$ by a scalar $\alpha \in \mathbb{R}_+$ to be the point which lies at distance $\alpha d(I, P)$ from $I$ in the direction of $P$, where $d(\cdot, \cdot)$ is the metric distance on $\mathrm{SPD}_n$. Geometrically, this is a transfer of the vector-space scalar multiplication on the tangent space to $\mathrm{SPD}_n$:

$$\alpha \otimes P = P^\alpha = \exp(\alpha \log(P)), \tag{3}$$

where $\exp, \log$ are the matrix exponential and logarithm. We further generalize the notion of scalar multiplication to allow for different relative expansion rates in different directions. For a fixed basepoint $I$ and a point $P \in \mathrm{SPD}_n$, we can replace the scalar $\alpha$ from Eq. 3 with an arbitrary real symmetric matrix $A \in S_n$. We define this *matrix scaling* by:

$$A \otimes P = \exp(A \odot \log(P)) \tag{4}$$

where $A \odot X$ denotes the Hadamard product. We denote the matrix scaling with $\otimes$, extending the previous usage: for any $\alpha \in \mathbb{R}$, we have $[\alpha] \otimes P = \alpha \otimes P$ where $[\alpha]$ is the matrix with every entry $\alpha$.

## 5 Implementation

In this section we detail how we learn representations in $\mathrm{SPD}_n$, and implement different linear mappings so that they conform to the premises of each operator, yielding SPD neural layers.

**Embeddings in** $\mathrm{SPD}_n$ **and** $S_n$**:** We are interested in learning embeddings in $\mathrm{SPD}_n$. To do so we exploit the connection between $\mathrm{SPD}_n$ and its tangent space $S_n$ through the exponential and logarithmic maps. To learn a point $P \in \mathrm{SPD}_n$, we first model it as a symmetric matrix $U \in S_n$. We impose symmetry on $U$ by learning a triangular matrix $X \in \mathbb{R}^{n \times n}$ with $n(n+1)/2$ parameters, such that $U = X + X^T$. To obtain the matrix $P \in \mathrm{SPD}_n$, we employ the exponential map: $P = \exp(U)$. Modeling points on the tangent space offers advantages for optimization, explained in §5. For the matrix scaling $A \otimes P$, we impose symmetry on the factor matrix $A \in S_n$ in the same way that we learn the symmetric matrix $U$.

**Isometries: Rotations and Reflections:** Rotations in $n$ dimensions are described as collections of pairwise orthogonal 2-dimensional rotations in planes (with a leftover 1-dimensional "axis of rotation" in odd dimensions). We utilize this observation to efficiently build elements of $O(n)$ out of two-dimensional rotations in coordinate planes. More precisely, for any $\theta \in [0, 2\pi]$ and choice of sign $\{+, -\}$ we let $R^\pm(\theta)$ denote the 2-dimensional rotation (+) or reflection (−) as $R^\pm(\theta) = \left( \begin{smallmatrix} \cos\theta & \mp\sin\theta \\ \sin\theta & \pm\cos\theta \end{smallmatrix} \right)$.

Then for any pair $i < j$ in $1 \ldots n$, we denote by $R_{ij}^\pm(\theta)$ the transformation which applies $R^\pm(\theta)$ to the $x_i x_j$-plane of $\mathbb{R}^n$, and leaves all other coordinates fixed. For example, in $O(5)$ the element $R_{24}^+(\theta)$ (see on the right) denotes the transformation where we replace the entries $(ii, ij, ji, jj)$ of $I_n$ with the corresponding values of $R^+(\theta)$. More general, rotations and reflections are built by taking products of these basic transformations. Given a $n(n-1)/2$-dimensional vector of angles $\vec{\theta} = (\theta_{12}, \ldots, \theta_{ij}, \ldots,)$

$$R_{24}^+(\theta) = \begin{pmatrix} 1 & 0 & 0 & 0 & 0 \\ 0 & \cos\theta & 0 & -\sin\theta & 0 \\ 0 & 0 & 1 & 0 & 0 \\ 0 & \sin\theta & 0 & \cos\theta & 0 \\ 0 & 0 & 0 & 0 & 1 \end{pmatrix}$$

and a choice of sign, we define the rotation and reflection corresponding to $\vec{\theta}$ by:

$$\text{Rot}(\vec{\theta}) = \prod_{i<j} R_{ij}^+(\theta_{ij}) \qquad \text{Refl}(\vec{\theta}) = \prod_{i<j} R_{ij}^-(\theta_{ij}) \tag{5}$$

where $\text{Rot}(\vec{\theta}), \text{Ref}(\vec{\theta}) \in \mathbb{R}^{n \times n}$ are the isometry matrices, and the vector of angles $\vec{\theta}$ can be regarded as a learnable parameter of the model. Finally, we denote the application of the transformation $M$ to the point $P \in \text{SPD}_n$ by:

$$M \odot P = MPM^T \tag{6}$$

**Optimization:** For the proposed rotations and reflections, the learnable weights are vectors of angles $\vec{\theta} \in \mathbb{R}^{\frac{n(n-1)}{2}}$, which do not pose an optimization challenge. On the other hand, embeddings in SPD have to be optimized respecting the geometry of the manifold, but as already explained, we model them on the space of symmetric matrices $S_n$, and then we apply the exponential map. In this manner, we are able to perform tangent space optimization [26] using standard Euclidean techniques, and circumvent the need for Riemannian optimization [15, 11], which we found to be less numerically stable. Due to the geometry of $\text{SPD}_n$ (see Appendix C.3), this is an exact procedure, which does not incur losses in representational power.

**Complexity:** The most frequently utilized operation when learning embeddings is the distance calculation, thus we analyze its complexity. In Appendix A.1 we detail the complexity of different operations. Calculating the distance between two points in $\text{SPD}_n$ implies computing multiplications, inversions and diagonalizations of $n \times n$ matrices. We find that the cost of the distance computation with respect to the matrix dimensions is $\mathcal{O}(n^3)$. Although a matrix of rank $n$ implies $n(n+1)/2$ dimensions thus a large $n$ value is usually not required, the cost of many operations is polynomial instead of linear.

**Towards neural network architectures:** We employ the proposed mappings along with the gyro-vector operations as building blocks for SPD neural layers. This is showcased in the experiments presented below. Scalings, rotations and reflections can be seen as feature transformations. Moreover, gyro-addition allows us to define the equivalent of bias addition. Finally, although we do not employ non-linearities, our approach can be seamlessly integrated with the ReEig layer (adaptation of a ReLU layer for SPD) proposed in [40].

## 6 Experiments

In this section we employ the transformations developed on SPD to build neural models for knowledge graph completion, item recommendation and question answering. Task-specific models in different geometries have been developed in the three cases, hence we consider them adequate benchmarks for representation learning.[3]

### 6.1 Knowledge Graph Completion

Knowledge graphs (KG) represent heterogeneous knowledge in the shape of *(head, relation, tail)* triples, where *head* and *tail* are entities and *relation* represents a relationships among entities. KG exhibit an intricate and varying structure where entities can be connected by symmetric, anti-symmetric, or hierarchical relations. To capture these non-trivial patterns more expressive modelling techniques become necessary [25], thus we choose this application to showcase the capabilities of our transformations on SPD manifolds. Given an incomplete KG, the task is to predict which unknown links are valid.

**Problem formulation:** Let $\mathcal{G} = (\mathcal{E}, \mathcal{R}, \mathcal{T})$ be a knowledge graph where $\mathcal{E}$ is the set of entities, $\mathcal{R}$ is the set of relations and $\mathcal{T} \subset \mathcal{E} \times \mathcal{R} \times \mathcal{E}$ is the set of triples stored in the graph. The usual approach is to learn a scoring function $\phi : \mathcal{E} \times \mathcal{R} \times \mathcal{E} \rightarrow \mathbb{R}$ that measures the likelihood of a triple to be true, with the goal of scoring all missing triples correctly. To do so, we propose to learn representations of entities as embeddings in $\text{SPD}_n$, and relation-specific transformation in the manifold, such that the KG structure is preserved.

---

[3]Code available at `https://github.com/fedelopez77/gyrospd`

**Scaling model:** We follow the base hyperbolic model MuRP [8] and adapt it into $\mathrm{SPD}_n$ by means of the *matrix scaling*. Its scoring function has shown success in the task given that it combines multiplicative and additive components, which are fundamental to model different properties of KG relations [4]. We translate it into $\mathrm{SPD}_n$ as:

$$\phi(h, r, t) = -d((\mathbf{M}_r \otimes \mathbf{H}) \oplus \mathbf{R}, \mathbf{T})^2 + b_h + b_t \tag{7}$$

where $\mathbf{H}, \mathbf{T} \in \mathrm{SPD}_n$ are embeddings and $b_h, b_t \in \mathbb{R}$ are scalar biases for the head and tail entities respectively. $\mathbf{R} \in \mathrm{SPD}_n$ and $\mathbf{M}_r$ are matrices that depend on the relation. For $d(\cdot, \cdot)$, we experiment with the Riemannian and the Finsler One metric distances.

**Isometric model:** A possible alternative is to embed the relation-specific transformations as elements of the $O(n)$ group (*i.e.*, rotations and reflections). This technique has proven effective in different metric spaces [25, 88]. In this case, $\mathbf{M}_r$ is a rotation or reflection matrix as in Eq. 5, and the scoring function is defined as:

$$\phi(h, r, t) = -d((\mathbf{M}_r \odot \mathbf{H}) \oplus \mathbf{R}, \mathbf{T})^2 + b_h + b_t \tag{8}$$

**Datasets:** We employ two standard benchmarks, namely WN18RR [16, 28] and FB15k-237 [16, 76]. WN18RR is a subset of WordNet [59] containing 11 lexical relationships between $40,943$ word senses. FB15k-237 is a subset of Freebase [14], with $14,541$ entities and $237$ relationships.

**Training:** We follow the standard data augmentation protocol by adding inverse relations to the datasets [48]. We optimize the cross-entropy loss with uniform negative sampling defined in Equation 9, where

$$\mathcal{L} = \sum_{(h,r,t)\in\mathcal{T}} \log(1 + \exp(Y_t \phi(h, r, t))) \tag{9}$$

$\mathcal{T}$ is the set of training triples, and $Y_t = -1$ if $t$ is a factual triple or $Y_t = 1$ if $t$ is a negative sample. We employ the AdamW optimizer [55]. We conduct a grid search with matrices of dimension $n \times n$ where $n \in \{14, 20, 24\}$ (this is the equivalent of $\{105, 210, 300\}$ degrees of freedom respectively) to select optimal dimensions, learning rate and weight decay, using the validation set. More details and set of hyperparameters in Appendix B.1.

**Evaluation metrics:** At test time, we rank the correct tail or head entity against all possible entities using the scoring function, and use inverse relations for head prediction [48]. Following previous work, we compute two ranking-based metrics: mean reciprocal rank (MRR), which measures the mean of inverse ranks assigned to correct entities, and hits at K (H@K, $K \in \{1, 3, 10\}$), which measures the proportion of correct triples among the top K predicted triples. We follow the standard evaluation protocol of filtering out all true triples in the KG during evaluation [16].

**Baselines:** We compare our models with their respective equivalents in different metric spaces, which are also state-of-the-art models for the task. For the scaling model, these are MuRE and MuRP [8], which perform the scaling operation in Euclidean and hyperbolic space respectively. For the isometric models, we compare to RotC [73], RotE and RotH, [25] (rotations in Complex, Euclidean and hyperbolic space respectively), and RefE and RefH [25] (reflections in Euclidean and hyperbolic space). Baseline results are taken from the original papers. We do not compare to previous work on SPD given that they lack the definition of an arithmetic operation in the space, thus a vis-a-vis comparison is not possible.

**Results:** We report the performance for all analyzed models, segregated by operation, in Table 1. On both dataset, the scaling model $\mathrm{SPD}_{\mathrm{Sca}}$ outperforms its direct competitors MuRE and MuRP, and this is specially notable in HR@10 for WN18RR: 59.0 for $\mathrm{SPD}_{\mathrm{Sca}}^{F_1}$ vs 55.4 and 56.6 respectively. SPD reflections are very effective on WN18RR as well. They outperform their Euclidean and hyperbolic counterparts RefE and RefH, in particular when equipped with the Finsler metric. Rotations on the SPD manifold, on the other hand, seem to be less effective. However, Euclidean and hyperbolic rotations require 500 dimensions whereas the $\mathrm{SPD}_{\mathrm{Rot}}$ models are trained on matrices of rank 14 (equivalent to 105 dims). Moreover, the underperformance observed in some of the analyzed cases for rotations and reflections does not repeat in the following experiments (§6.2 & §6.3). Hence, we consider this is due to overfitting in some particular setups. Although we tried different regularization methods, we regard a sub-optimal configuration rather than a geometric reason to be the cause for the underperformance.

Regarding the choice of a distance metric, the Finsler One metric is better suited with respect to HR@3 and HR@10 when using scalings and reflections on WN18RR. For the FB15k-237 dataset, SPD models operating with the Riemannian metric outperform their Finsler counterparts. This

Table 1: Results for Knowledge graph completion.

| Operation | Model | WN18RR | | | | FB15k-237 | | | |
|---|---|---|---|---|---|---|---|---|---|
| | | MRR | HR@1 | HR@3 | HR@10 | MRR | HR@1 | HR@3 | HR@10 |
| Scaling | MuRE | 47.5 | 43.6 | 48.7 | 55.4 | 33.6 | 24.5 | 37.0 | 52.1 |
| | MuRP | 48.1 | **44.0** | 49.5 | 56.6 | 33.5 | 24.3 | 36.7 | 51.8 |
| | $\text{SPD}_{\text{Sca}}^{R}$ | 48.1 | 43.1 | 50.1 | 57.6 | **34.5** | **25.1** | **38.0** | **53.5** |
| | $\text{SPD}_{\text{Sca}}^{F_1}$ | **48.4** | 42.6 | **51.0** | **59.0** | 32.9 | 23.6 | 36.3 | 51.5 |
| Rotations | RotC | 47.6 | 42.8 | 49.2 | 57.1 | 33.8 | 24.1 | 37.5 | 53.3 |
| | RotE | 49.4 | 44.6 | 51.2 | 58.5 | **34.6** | **25.1** | **38.1** | **53.8** |
| | RotH | **49.6** | **44.9** | **51.4** | **58.6** | 34.4 | 24.6 | 38.0 | 53.5 |
| | $\text{SPD}_{\text{Rot}}^{R}$ | 46.2 | 39.7 | 49.6 | 57.8 | 32.9 | 23.6 | 36.3 | 51.6 |
| | $\text{SPD}_{\text{Rot}}^{F_1}$ | 40.9 | 30.5 | 48.2 | 57.3 | 32.1 | 22.9 | 35.4 | 50.5 |
| Reflections | RefE | 47.3 | 43.0 | 48.5 | 56.1 | **35.1** | **25.6** | **39.0** | **54.1** |
| | RefH | 46.1 | 40.4 | 48.5 | 56.8 | 34.6 | 25.2 | 38.3 | 53.6 |
| | $\text{SPD}_{\text{Ref}}^{R}$ | 48.3 | 44.0 | 49.7 | 56.7 | 32.5 | 23.4 | 35.6 | 51.0 |
| | $\text{SPD}_{\text{Ref}}^{F_1}$ | **48.7** | **44.3** | **50.1** | **57.4** | 31.6 | 22.5 | 34.6 | 50.0 |

suggests that the Riemannian metric is capable of disentangling the large number of relationships in this dataset to a better extent.

In these experiments we have evaluated models applying equivalent operations and scoring functions in different geometries, thus they can be thought as a vis-a-vis comparison of the metric spaces. We observe that SPD models tie or outperform baselines in most instances. This showcases the improved representation capacity of the SPD manifold when compared to Euclidean and hyperbolic spaces. Moreover, it demonstrates the effectiveness of the proposed metrics and operations in this manifold.

## 6.2 Knowledge Graph Recommender Systems

Recommender systems (RS) model user preferences to provide personalized recommendations [93]. KG embedding methods have been widely adopted into the recommendation problem as an effective tool to model side information and enhance the performance [91, 34]. For instance, one reason for recommending a movie to a particular user is that the user has already watched many movies from the same genre or director [56]. Given multiple relations between users, items, and heterogeneous entities, the goal is to predict the user's next item purchase or preference.

**Model:** We model the recommendation problem as a link prediction task over users and items [49]. In addition, we aim to incorporate side information between users, items and other entities. Hence we apply our KG embedding method from §6.1 as is, to embed this multi-relational graph. We evaluate the capabilities of the approach by only measuring the performance over user-item interactions.

**Datasets:** To investigate the recommendation problem endowed with added relationships, we employ the Amazon dataset [58, 61] (branches "Software", "Luxury & Beauty" and "Prime Pantry"), with users' purchases of products, and the MindReader dataset [18] of movie recommendations. Both datasets provide additional relationships between users, items and entities such as product brands, or movie directors and actors. To generate evaluation splits, the penultimate and last item the user has interacted with are withheld as dev and test sets respectively.

**Training:** In this setup we also augment the data by adding inverse relations and optimize the loss from Equation 9. We set the size of the matrices to $10 \times 10$ dimensions (equivalent to 55 free parameters). More details about relationships and set of hyperparameters in Appendix B.2.

**Evaluation and metrics:** We follow the standard procedure of evaluating against 100 randomly selected samples the user has not interacted with [38, 53]. To evaluate the recommendation performance we focus on the *buys / likes* relation. For each user $u$ we rank the items $i_j$ according to the scoring function $\phi(u, buys, i_j)$. We adopt MRR and H@10, as ranking metrics for recommendations.

**Baselines:** We compare to TransE [16], RotC [73], MuRE and MuRP [8] trained with 55 dimensions.

**Results:** In Table 2 we observe that the SPD models tie or outperform the baselines in both MRR and HR@10 across all analyzed datasets. Rotations in both, Riemannian and Finsler metrics, are more effective in this task, achieving the best performance in 3 out of 4 cases, followed by the scaling models. Overall, this shows the capabilities of the systems to effectively represent user-item interactions enriched with relations between items and their attributes, thus learning to better model users' preferences. Furthermore, it displays the versatility of the approach to diverse data domains.

Table 2: Results for Knowledge graph-based recommender systems.

| Model | SOFTWARE | | LUXURY | | PANTRY | | MINDREADER | |
|---|---|---|---|---|---|---|---|---|
| | MRR | H@10 | MRR | H@10 | MRR | H@10 | MRR | H@10 |
| TRANSE | 28.5±0.1 | 47.2±0.5 | 35.6±0.1 | 52.3±0.1 | 16.6±0.0 | 35.3±0.1 | 19.1±0.4 | 37.6±0.1 |
| ROTC | 28.5±0.3 | 45.4±1.4 | 33.0±0.1 | 49.8±0.2 | 14.5±0.0 | 31.3±0.2 | 25.3±0.3 | 50.3±0.6 |
| MURE | 29.4±0.4 | 47.1±0.4 | 35.6±0.7 | 54.0±0.3 | 19.4±0.1 | 39.5±0.2 | 25.2±0.3 | 49.9±0.6 |
| MURP | 29.6±0.3 | 47.9±0.3 | **37.5±0.1** | **55.2±0.3** | 19.4±0.1 | 39.8±0.2 | 25.3±0.3 | 49.3±0.2 |
| $\text{SPD}_{\text{Sca}}^{R}$ | 29.4±0.4 | 48.1±0.8 | **37.5±0.2** | 55.1±0.2 | 19.5±0.0 | 39.6±0.3 | 25.4±0.1 | 49.8±0.3 |
| $\text{SPD}_{\text{Sca}}^{F_1}$ | 28.8±0.1 | 46.9±0.5 | 37.3±0.3 | 54.1±0.9 | 19.0±0.1 | 38.8±0.2 | **25.7±0.5** | 49.5±0.1 |
| $\text{SPD}_{\text{Rot}}^{R}$ | **30.3±0.2** | 48.6±0.9 | 37.2±0.1 | 54.8±0.4 | **20.0±0.1** | **40.3±0.1** | 25.3±0.0 | **50.5±0.3** |
| $\text{SPD}_{\text{Rot}}^{F_1}$ | 30.1±0.1 | **49.1±0.3** | 36.9±0.1 | 54.5±0.6 | 19.2±0.0 | 39.3±0.1 | **25.7±0.0** | 49.5±0.2 |
| $\text{SPD}_{\text{Ref}}^{R}$ | 29.6±0.2 | 48.0±0.5 | 37.3±0.2 | 55.0±0.2 | 19.3±0.0 | 39.7±0.3 | 25.3±0.0 | 49.1±0.1 |
| $\text{SPD}_{\text{Ref}}^{F_1}$ | 29.3±0.1 | 47.5±0.6 | 36.8±0.0 | 54.8±0.1 | 18.6±0.2 | 38.3±0.3 | 24.8±0.2 | 47.9±1.8 |

## 6.3 Question Answering

We evaluate our approach on the task of Question Answering (QA). In this manner we also showcase the capabilities of our methods to train word embeddings.

**Model:** We adapt the model from HyperQA [74] to SPD. We model word embeddings $t_i \in \text{SPD}_n$, and represent question/answers as a summation of the embeddings of their corresponding tokens. We apply a feature transformation $T(\cdot)$ followed by a bias addition, as an equivalent of a neural linear layer. $T(\cdot)$ can be a scaling, rotation or reflec-

$$\text{sim}(q, a) = -w_f d(\mathbf{Q}, \mathbf{A}) + w_b,$$
$$\text{where } \mathbf{Q} = T(\bigoplus_{i=1}^{n} t_i^q) \oplus B \quad (10)$$

tion. Finally we compute a distance-based similarity function between the resulting question/answer representations as defined in Equation 10, where $w_f, w_b \in \mathbb{R}$, $B \in \text{SPD}_n$ and the transformation $T$ are parameters of the model.

**Datasets:** We analyze two popular benchmarks for QA: TrecQA [85] (clean version) and WikiQA [89], filtering out questions with multiple answers from the dev and test sets.

**Training:** We optimize the cross-entropy loss from Eq. 9, where we replace $\phi$ for $\text{sim}(q, a)$ and for each question we use wrong answers as negative samples. We set the size of the matrices to $14 \times 14$ dimensions (equivalent to 105 free parameters). The set of hyperparameters can be found in Appendix B.3.

**Evaluation metrics:** At test time, for each question we rank its candidate answers according to Eq. 10. We adopt MRR and H@1 as evaluation metrics.

Table 3: Results for Question Answering.

| Model | TRECQA | | WIKIQA | |
|---|---|---|---|---|
| | MRR | H@1 | MRR | H@1 |
| Euclidean | 55.9±2.0 | 41.0±2.0 | 43.4±0.3 | 22.4±1.1 |
| Hyperbolic | 58.0±1.3 | 39.3±2.0 | 44.0±0.4 | 22.8±0.6 |
| $\text{SPD}_{\text{Sca}}^{R}$ | 55.4±0.1 | 37.1±0.1 | **45.5±0.5** | 24.4±1.1 |
| $\text{SPD}_{\text{Sca}}^{F_1}$ | 57.1±0.7 | 38.6±0.2 | 44.8±0.5 | 24.0±0.6 |
| $\text{SPD}_{\text{Rot}}^{R}$ | 58.7±1.5 | 41.4±2.9 | 44.6±0.6 | 23.6±0.6 |
| $\text{SPD}_{\text{Rot}}^{F_1}$ | 58.1±0.5 | **43.6±1.0** | 43.7±0.4 | 23.8±0.8 |
| $\text{SPD}_{\text{Ref}}^{R}$ | 57.3±0.3 | 40.7±1.1 | 43.9±0.7 | 23.4±2.0 |
| $\text{SPD}_{\text{Ref}}^{F_1}$ | **59.6±0.5** | 42.1±1.0 | 44.7±1.2 | **25.0±2.5** |

**Baselines:** We compare against Euclidean and hyperbolic spaces of 105 dimensions. For the Euclidean model we employ a linear layer as feature transformation. For the hyperbolic model, we operate on the tangent space and project the points into the Poincaré ball to compute distances.

**Results:** We present results in Table 3. In both datasets, we see that the word embeddings and transformations learned by the SPD models are able to place questions and answers representations in the space such that they outperform Euclidean and hyperbolic baselines. Finsler metrics seem to be very effective in this scenario, improving the performance of their Riemannian counterparts in many cases. Overall, this suggests that embeddings in SPD manifolds learn meaningful representations that can be exploited into downstream tasks. Moreover, we showcase how to employ different operations as feature transformations and bias additions, replicating the behavior of linear layers in classical deep learning architectures that can be seamlessly integrated with different distance metrics.

## 6.4 Analysis

One reason to embed data into Riemannian manifolds, such as SPD, is to use geometric properties of the manifold to analyze the structure of the data [52]. Visualizations in SPD are difficult due to their

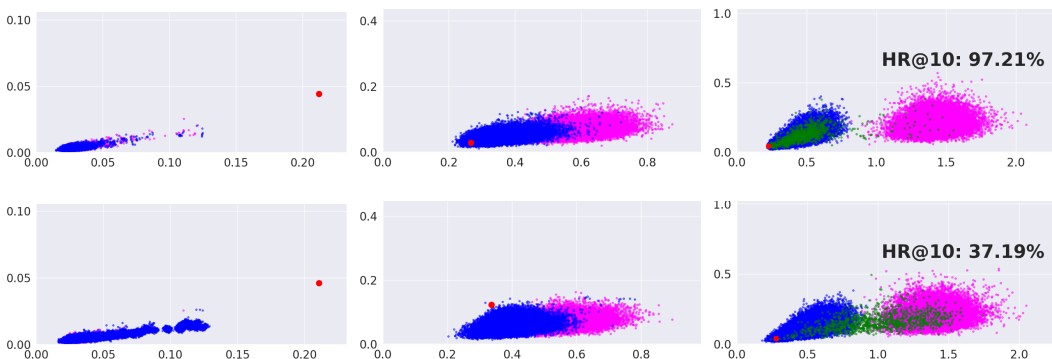

Figure 5: Train, negative and validation triples for relationships *'derivationally related form'* (top) and *'hypernym'* (bottom) for 5 (left), 50 (center) and 3000 (right) epochs for the $\text{SPD}^{F_1}_{\text{Sca}}$ model. The red dot corresponds to the relation addition **R**.

high dimensionality. As a solution we use the vector-valued distance function to develop a new tool to visualize and analyze structural properties of the learned representations.

We adopt the vector $(n-1, n-3, \cdots, -n+3, -n+1)$, as the barycenter of the space in $\mathbb{R}^n$ where the VVD is contained. Then, we plot the norm of the VVD vector and its angle with respect to this barycenter. In Figure 5, we compute and plot the VVD corresponding to $d(\mathbf{M}_r \otimes \mathbf{H}, \mathbf{T})$ and **R** as defined in Eq. 7 for KG models trained on WN18RR. In early stages of the training, all points fall near the origin (left side of the plots). As training evolves, the model learns to separate true $(h, r, t)$ triples from corrupted ones (center part). When the training converges, the model is able to clearly disentangle and cluster positive and negative samples. We observe how the position of the validation triples (green points, not seen during training) directly correlates with the performance of each relation. Plots for more relations in Appendix B.1.

## 7   Conclusions

Riemannian geometry has gained attention due to its capacity to represent non-Euclidean data arising in several domains. In this work we introduce the vector-valued distance function, which allows to implement universal models (generalizing previous metrics on SPD), and can be leveraged to provide a geometric interpretation on what the models learn. Moreover, we bridge the gap between Euclidean and SPD geometry under the lens of the gyrovector theory, providing means to transfer standard arithmetic operations from the Euclidean setting to their analog notions in SPD. These tools enable practitioners to exploit the full representation power of SPD, and profit from the enhanced expressivity of this manifold. We propose and evaluate SPD models on three tasks and eight datasets, which showcases the versatility of the approach and ease of integration with downstream tasks. The results reflect the superior expressivity of SPD when compared to Euclidean or hyperbolic baselines.

This work is not without limitations. We consider the computational complexity of working with spaces of matrices to be the main drawback, since the cost of many operations is polynomial instead of linear. Nevertheless, a matrix of rank $n$ implies $n(n + 1)/2$ dimensions thus a large $n$ value is usually not required.

## Acknowledgments and Disclosure of Funding

This work has been supported by the German Research Foundation (DFG) as part of the Research Training Group Adaptive Preparation of Information from Heterogeneous Sources (AIPHES) under grant No. GRK 1994/1, as well as under Germany's Excellence Strategy EXC-2181/1 - 390900948 (the Heidelberg STRUCTURES Cluster of Excellence), and by the Klaus Tschira Foundation, Heidelberg, Germany.

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
