# A  Implementation Details

## A.1  Computational Complexity of Operations

In this section we discuss the computational theoretical complexity of the different operations involved in the development of this work. We employ Big O notation[4]. Since in all cases operations are not nested, but are applied sequentially, the costs can be added resulting in a polynomial expression. Thus, by applying the properties of the notation, we disregard lower-order terms of the polynomial.

**Matrix Operations:**  For $n \times n$ matrices, the associated complexity of each operation is as follows:[5]

- Addition and subtraction: $\mathcal{O}(n^2)$
- Multiplication: $\mathcal{O}(n^{2.4})$
- Inversion: $\mathcal{O}(n^{2.4})$
- Diagonalization: $\mathcal{O}(n^3)$

**SPD Operations:**  For $n \times n$ SPD matrices, the associated complexity of each operation is as follows:

- Exp/Log map: $\mathcal{O}(n^3)$, due to diagonalizations.
- Gyro-Addition: $\mathcal{O}(n^{2.4})$, due to matrix multiplications
- Matrix Scaling: $\mathcal{O}(n^3)$, due to exp and log maps.
- Isometries: $\mathcal{O}(n^{2.4})$, due to matrix multiplications.

**Distance Calculation:**  The full computation of the distance algorithm in $\mathrm{SPD}_n$ involves matrix square root, inverses, multiplications, and diagonalizations. Since they are applied sequentially, without affecting the dimensionality of the matrices, we can take the highest value as the asymptotic cost of the algorithm, which is $\mathcal{O}(n^3)$.

## A.2  Tangent Space Optimization

Optimization in Riemannian manifolds normally requires Riemannian Stochastic Gradient Descent (RSGD) [15] or other Riemannian techniques [11]. We performed initial tests converting the Euclidean gradient into its Riemannian form, but found it to be less numerically stable and also slower than tangent space optimization [26]. With tangent space optimization, we can use standard Euclidean optimization techniques, and respect the geometry of the manifold. Note that tangent space optimization is an exact procedure, which does not incur losses in representational power. This is the case in $\mathrm{SPD}_n$ specifically because of a completeness property given by the choice of $I \in \mathrm{SPD}_n$ as the basepoint: there is always a global bijection between the tangent space and the manifold.

# B  Experimental Details

All models and experiments were implemented in PyTorch [64] with distributed data parallelism, for high performance on clusters of CPUs/GPUs.

**Hardware:**  All experiments were run on Intel Cascade Lake CPUs, with microprocessors Intel Xeon Gold 6230 (20 Cores, 40 Threads, 2.1 GHz, 28MB Cache, 125W TDP). Although the code supports GPUs, we did not utilize them due to higher availability of CPU's.

---

[4]`https://en.wikipedia.org/wiki/Big_O_notation`
[5]`https://en.wikipedia.org/wiki/Computational_complexity_of_mathematical_operations`

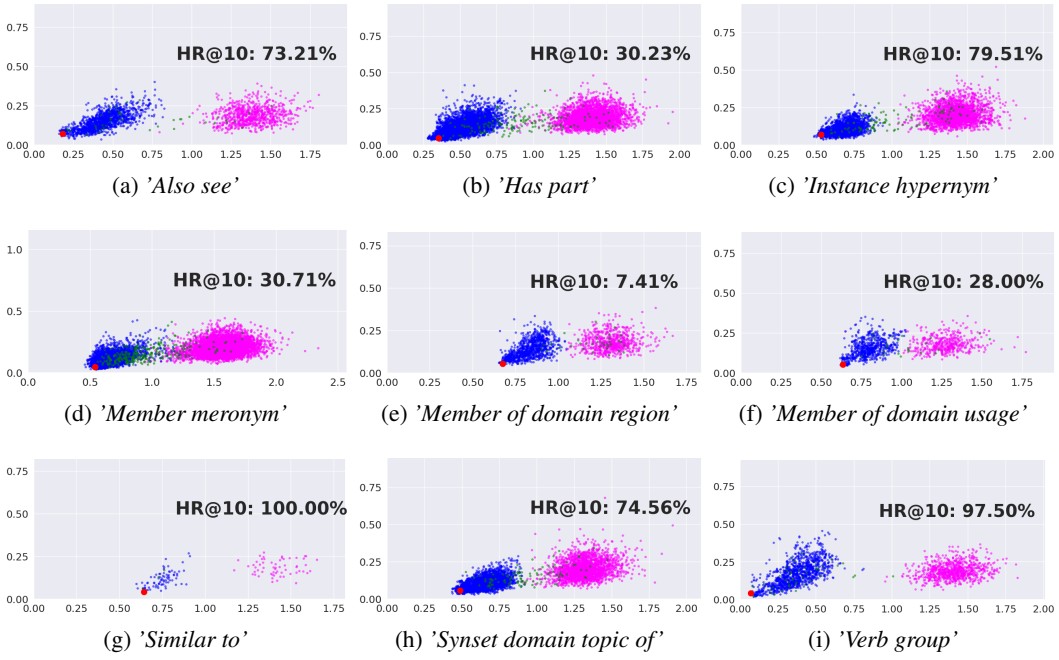

Figure 6: Train, negative and validation triples for WN18RR relationships of the $\mathrm{SPD}_{\mathrm{Sca}}^{F_1}$ model after convergence. The red dot corresponds to the relation addition **R**.

## B.1 Knowledge Graph Completion

**Setup:** We train for $5000$ epochs, with batch size of $4096$ and $10$ negative samples, reducing the learning rate by a factor of $2$ if the model does not improve the performance on the dev set after $50$ epochs, and early stopping based on the MRR if the model does not improve after $500$ epochs. We use the burn-in strategy [62] training with a $10$ times smaller learning rate for the first $10$ epochs. We experiment with matrices of dimension $n \times n$ where $n \in \{14, 20, 24\}$ (this is the equivalent of $\{105, 210, 300\}$ degrees of freedom respectively), learning rates from $\{1e{-}4, 5e{-}5, 1e{-}5\}$ and weight decays of $\{1e{-}2, 1e{-}3\}$.

**Datasets:** Stats about the datasets used in Knowledge graph experiments can be found in Table 4.

**Results:** In addition to the results provided in §6.1, in Table 5 we provide a comparison with other state-of-the-art models. We include ComplEx [77], Tucker [9], and Quaternion [92].

**Analisis:** In Figure 6 we add equivalent plots to the ones explained in §6.4 for other relations from WN18RR.

## B.2 Knowledge Graph Recommender Systems

**Setup:** We train for $3000$ epochs, with batch size from $\{512, 1024\}$ and $10$ negative samples, and early stopping based on the MRR if the model does not improve after $200$ epochs. We use the burn-in strategy [62] training with a $10$ times smaller learning rate for the first $10$ epochs. We report average $\pm$ standard deviation of $3$ runs. We experiment with matrices of dimension $10 \times 10$ (this is the

Table 4: Statistics for Knowledge Graph completion datasets.

| Dataset | Entities | Relations | Train | Dev | Test |
|---------|----------|-----------|-------|-----|------|
| WN18RR | 40943 | 11 | 86835 | 3034 | 3134 |
| FB15k-237 | 14541 | 237 | 272115 | 17535 | 20466 |

Table 5: Results for Knowledge graph completion.

| Space | Model | WN18RR | | | | FB15k-237 | | | |
|---|---|---|---|---|---|---|---|---|---|
| | | MRR | HR@1 | HR@3 | HR@10 | MRR | HR@1 | HR@3 | HR@10 |
| $\mathbb{C}$ | ComplEx | 48.0 | 43.5 | 49.5 | 57.2 | 35.7 | 26.4 | 39.2 | 54.7 |
| | RotC | 47.6 | 42.8 | 49.2 | 57.1 | 33.8 | 24.1 | 37.5 | 53.3 |
| $\mathbb{Q}$ | Quaternion | 48.8 | 43.8 | 50.8 | 58.2 | **36.6** | **27.1** | **40.1** | **55.6** |
| $\mathbb{R}$ | Tucker | 47.0 | 44.3 | 48.2 | 52.6 | 35.8 | 26.6 | 39.4 | 54.4 |
| | MuRE | 47.5 | 43.6 | 48.7 | 55.4 | 33.6 | 24.5 | 37.0 | 52.1 |
| | RotE | 49.4 | 44.6 | 51.2 | 58.5 | 34.6 | 25.1 | 38.1 | 53.8 |
| | RefE | 47.3 | 43.0 | 48.5 | 56.1 | 35.1 | 25.6 | 39.0 | 54.1 |
| $\mathbb{H}$ | MuRP | 48.1 | 44.0 | 49.5 | 56.6 | 33.5 | 24.3 | 36.7 | 51.8 |
| | RotH | **49.6** | **44.9** | **51.4** | 58.6 | 34.4 | 24.6 | 38.0 | 53.5 |
| | RefH | 46.1 | 40.4 | 48.5 | 56.8 | 34.6 | 25.2 | 38.3 | 53.6 |
| SPD | $\text{SPD}_{\text{Sca}}^{R}$ | 48.1 | 43.1 | 50.1 | 57.6 | 34.5 | 25.1 | 38.0 | 53.5 |
| | $\text{SPD}_{\text{Sca}}^{F1}$ | 48.4 | 42.6 | 51.0 | **59.0** | 32.9 | 23.6 | 36.3 | 51.5 |
| | $\text{SPD}_{\text{Rot}}^{R}$ | 46.2 | 39.7 | 49.6 | 57.8 | 32.7 | 23.4 | 36.1 | 51.4 |
| | $\text{SPD}_{\text{Rot}}^{F1}$ | 38.6 | 25.6 | 48.7 | 56.8 | 31.4 | 22.3 | 34.7 | 49.8 |
| | $\text{SPD}_{\text{Ref}}^{R}$ | 48.3 | 44.0 | 49.7 | 56.7 | 32.5 | 23.4 | 35.6 | 51.0 |
| | $\text{SPD}_{\text{Ref}}^{F1}$ | 48.7 | 44.3 | 50.1 | 57.4 | 30.7 | 21.7 | 33.7 | 48.8 |

equivalent of $\{55\}$ degrees of freedom), learning rates from $\{5e-4, 1e-4, 5e-5\}$ and weight decay of $1e-3$. Same grid search is applied to baselines.

**Datasets:** On the Amazon dataset we adopt the 5-core split for the branches "Software", "Luxury & Beauty" and "Industrial & Scientific", which form a diverse dataset in size and domain. We add relationships used in previous work [95, 3]. These are:

- *also_bought*: users who bought item A also bought item B.
- *also_view*: users who bought item A also viewed item B.
- *category*: the item belongs to one or more categories.
- *brand*: the item belongs to one brand.

On the MindReader dataset, we consider a user-item interaction when a user gave an explicit positive rating to the movie. The relationships added are:

- *directed_by*: the movie was directed by this person.
- *produced_by*: the movie was produced by this person/company.
- *from_decade*: the movie was released in this decade.
- *followed_by*: the movie was followed by this other movie.
- *has_genre*: the movie belongs to this genre.
- *has_subject*: the movie has this subject.
- *starring*: the movie was starred by this person.

Statistics of the datasets with the added relationships can be seen in Table 6. For dev/test we only consider users with 3 or more interactions.

Table 6: Statistics for KG Recommender datasets.

| Dataset | Users | Items | Other Entities | Train Relations | | Dev/Test |
|---|---|---|---|---|---|---|
| | | | | User-item | Others | |
| Software | 1826 | 802 | 689 | 8242 | 6078 | 1821 |
| Luxury Beauty | 3819 | 1581 | 2 | 20796 | 26044 | 3639 |
| Prime Pantry | 14180 | 4970 | 1100 | 102848 | 99118 | 14133 |
| MindReader | 961 | 2128 | 11775 | 11279 | 99486 | 953 |

Table 7: Statistics for Question Answering datasets.

|  | TRECQA | WIKIQA |
| --- | --- | --- |
| Train Qs | 1227 | 2119 |
| Dev Qs | 65 | 127 |
| Test Qs | 68 | 244 |
| Train pairs | 53417 | 20361 |
| Dev Pairs | 1117 | 1131 |
| Test Pairs | 1442 | 2352 |

### B.3 Question Answering

**Setup:** We train for 300 epochs, with 2 negative samples and early stopping based on the MRR if the model does not improve after 20 epochs. We use the burn-in strategy [62] training with a 10 times smaller learning rate for the first 10 epochs. We report average $\pm$ standard deviation of 3 runs. We experiment with matrices of dimension $14 \times 14$ (equivalent of 105 degrees of freedom respectively), batch size from $\{512, 1024\}$, learning rate from $\{1e-4, 5e-5, 1e-5\}$ and weight decays of $1e-3$. Same grid search was applied to baselines.

**Datasets:** Stats about the datasets used for Question Answering experiments can be found in Table 7.

## C Differential Geometry of $\mathrm{SPD}_n$

### C.1 Orthogonal Diagonalization

Every real symmetric matrix may be orthogonally diagonalized: For every point $P \in \mathrm{SPD}_n$ we may find a positive diagonal matrix $D$ and an orthogonal matrix $K$ such that $P = KDK^T$. This diagonalization has two practical consequences: it allows efficient computation of important $\mathrm{SPD}_n$ operations, and provides another means of generalizing Euclidean notions to $\mathrm{SPD}_n$.

With respect to computation, if $P \in \mathrm{SPD}_n$ has orthogonal diagonalization $P = KDK^T$, we may compute its square root and logarithm as $\sqrt{P} = K\sqrt{D}K^T$ and $\log(P) = K\log(D)K^T$ where $\sqrt{D} = \mathrm{diag}(\sqrt{d_1}, \ldots, \sqrt{d_n})$ and $\log(D) = \mathrm{diag}(\log d_1, \ldots, \log d_n)$ for $D = \mathrm{diag}(d_1, \ldots, d_n)$. Similarly, if a tangent vector $U \in S_n$ has orthogonal diagonalization $U = K\Lambda K^T$ (here $\Lambda = \mathrm{diag}(\lambda_1, \ldots, \lambda_n)$ not necessarily positive definite), the exponential map is computed as $\exp(U) = Ke^\Lambda K^T$, where $e^\Lambda = \mathrm{diag}(e^{\lambda_1}, \ldots e^{\lambda_n})$.

We verify this in the two lemmas below.

**Lemma 1.** *If $K \in O(n)$ and $X$ is any $n \times n$ matrix, then $\exp(KXK^T) = K\exp(X)K^T$.*

*Proof.* As $K$ is orthogonal, $K^T = K^{-1}$. Conjugation is an automorphism of the algebra of $n \times n$ matrices, and so applying this to any partial sum of the exponential $\exp(X) = \sum_{n=0}^{\infty} \frac{1}{n!}X^n$ yields

$$\sum_{n=0}^{N} \frac{1}{n!}(KXK^{-1})^n = K\left(\sum_{n=0}^{N} \frac{1}{n!}X^n\right)K^{-1}.$$

Taking the limit of this equality as $N \to \infty$ gives the claimed result. $\square$

**Lemma 2.** *If $D = \mathrm{diag}(d_1, \ldots, d_n)$ is a diagonal matrix, then $\exp(D) = \mathrm{diag}(e^{d_1}, \ldots, e^{d_n})$.*

*Proof.* The multiplication of diagonal matrices coincides with the elementwise product of their diagonal entries. Again applying this to any partial sum of the exponential of $D = \mathrm{diag}(d_1, \ldots, d_n)$ gives

$$\sum_{n=0}^{N} \frac{1}{n!}\mathrm{diag}(\ldots, d_i \ldots)^n = \mathrm{diag}\left(\ldots, \sum_{n=0}^{N} \frac{1}{n!}d_i^n, \ldots\right).$$

Taking the limit of this equality as $N \to \infty$ gives the claimed result. $\square$

## C.2 Metric and Isometries

The Riemannian metric on $\mathrm{SPD}_n$ is defined as follows: if $U, V \in S_n$ are tangent vectors based at $P \in \mathrm{SPD}_n$, their inner product is:

$$\langle U, V \rangle_P = \mathrm{tr}(P^{-1} U P^{-1} V).$$

Note that at the basepoint, this is just the standard matrix inner product $\langle U, V \rangle_I = \mathrm{tr}(UV^T)$ as $U, V$ are symmetric. We now verify the $GL(n, \mathbb{R})$ action given by $M$ acting as $P \mapsto MPM^T$ is an action by isometries of this metric.

**Lemma 3.** *The action $f(P) = MPM^T$ extends to tangent vectors $U$ based at $P$ without change in formula: $f_*(U) = MUM^T$*

*Proof.* Let $P \in \mathrm{SPD}_n$ and $U \in S_n$ be a tangent vector based at $P$. Then by definition, $U = P_0'$ is the derivative of some path $P_t$ of some path of matrices in $\mathrm{SPD}_n$ throguh $P_0 = P$. We compute the action of $P \to MPM^T$ on $U$ by taking the derivative of its action on the path:

$$\frac{d}{dt}\Big|_{t=0} MP_t M = MP_t' M\Big|_{t=0} = MUM^T$$

$\square$

**Proposition 1.** *For every $M \in GL(n; \mathbb{R})$ the transformation $M \mapsto MPM^T$ preserves the Riemannian metric on $\mathrm{SPD}_n$.*

*Proof.* Let $M \in GL(n; \mathbb{R})$ and choose arbitrary point $P \in \mathrm{SPD}_n$, and tangent vectors $U, V \in T_P \mathrm{SPD}_n$. We compute the pullback of the metric under the symmetry $f(P) = MPM^T$. Computing directly from the definition an the previous lemma,

$$
\begin{aligned}
f^*\langle U, V \rangle_P &= \langle f_* U, f_* V \rangle_{f(P)} \\
&= \langle MUM^T, MVM^T \rangle_{MPM^T} \\
&= \mathrm{tr}\left( \left(MPM^T\right)^{-1} MUM^T \left(MPM^T\right)^{-1} MVM^T \right) \\
&= \mathrm{tr}\left( M^{-T} P^{-1} U P^{-1} V M^T \right) \\
&= \mathrm{tr}\left( P^{-1} U P^{-1} V \right) \\
&= \langle U, V \rangle_P,
\end{aligned}
$$

where the penultimate equality uses that trace is invariant under conjugacy.

$\square$

This provides a vivid geometric interpretation of the previously discussed orthogonal diagonalization operation on $\mathrm{SPD}_n$.

**Corollary 1.** *Given any $P \in \mathrm{SPD}_n$, there exists a symmetry fixing $I$ which moves $P$ to a diagonal matrix.*

This subspace of diagonal matrices plays an essential role in working with $\mathrm{SPD}_n$. As we verify below, the intrinsic geometry of this subspace of diagonal matrices inherited from the Riemannian metric on $\mathrm{SPD}_n$ is flat.

**Proposition 2.** *Let $\mathcal{D} \subset \mathrm{SPD}_n$ be the set of diagonal matrices, and define $f: \mathbb{R}^n \to \mathcal{D}$ by $f(x_1, \ldots, x_n) = \mathrm{diag}(e^{x_1}, \ldots, e^{x_n})$. Then $f$ is an isometry from the Euclidean metric on $\mathbb{R}^n$ to the metric on $\mathcal{D}$ induced from $\mathrm{SPD}_n$.*

*Proof.* We pull back the metric on $\mathcal{D}$ by $f$, and see that on $\mathbb{R}^n$ this results in the standard Euclidean metric. Given a point $x \in \mathbb{R}^n$ with tangent vectors $y, z \in \mathbb{R}^n$, we compute this as

$$f^*\langle y, z \rangle_x = \langle f_* y, f_* z \rangle_{f(x)}$$

From the definition of $f$, we see that the pushforward of $y$ along $f$ is $\mathrm{diag}(\ldots, e^{x_i} y_i, \ldots)$ and similarly for $z$. Thus we may compute directly and see the result is the standard dot product on $\mathbb{R}^n$.

$$
\begin{aligned}
\langle \mathrm{diag}(e^{x_i} y_i), \mathrm{diag}(e^{x_i} z_i) \rangle_{\mathrm{diag}(e^{x_i})} &= \mathrm{tr}\left( \mathrm{diag}(e^{x_i} y_i) \, \mathrm{diag}(e^{-x_i}) \, \mathrm{diag}(e^{x_i} y_i) \, \mathrm{diag}(e^{-x_i}) \right) \\
&= \mathrm{tr}\left( \mathrm{diag}(y_i z_i) \right) \\
&= \sum_{i=1}^{n} y_i z_i
\end{aligned}
$$

$\square$

This subspace $\mathcal{D}$ is in fact a *maximal flat* for $\mathrm{SPD}_n$, the largest dimensional totally geodesic Euclicean submanifold embedded in $\mathrm{SPD}_n$. For more information on the general theory of symmetric spaces from which the notion of maximal flats arises, see Helgason [39]. For our purposes, it is only important to note the following fact.

**Corollary 2.** *The set of diagonal matrices in $\mathrm{SPD}_n$ is an isometrically and totally geodesically embedded copy of euclidean $n$-space.*

### C.3  Exponential and Logarithmic Maps

The Riemannian exponential map gives a connection between the Euclidean geometry of the tangent space $S_n$ and the curved geometry of $\mathrm{SPD}_n$. It assigns the tangent vector $U$ to the point $Q = \exp(U)$ of $\mathrm{SPD}_n$ reached by traveling along the geodesic starting from the basepoint $I$ in direction $U$ for distance $\|U\|$.

As a consequence of non-positive curvature, $\exp$ is a diffeomorphism of $S_n$ onto $\mathrm{SPD}_n$, and so has an inverse: the Riemannian logarithm $\log \colon \mathrm{SPD}_n \to S_n$. See [10] for a review of the general theory of manifolds of non-positive curvature. Together, this pair of functions allows one to freely move between 'tangent space coordinates' or the original 'manifold coordinates' which we exploit to transfer Euclidean optimization schemes to $\mathrm{SPD}_n$ (see §5).

Secondly, the geometry of $\mathrm{SPD}_n$ is so tightly tied to the algebra of $n \times n$ matrices that the Riemannian exponential agrees exactly with the usual matrix exponential, and the Riemannian logarithm is the matrix logarithm (because of this, we do not distinguish the two notationally), as we verify in the proposition below. Both of these are readily computable via orthogonal diagonalization, as given in §C.1. This is in stark contrast to general Riemannian manifolds, where the exponential map may have no simple formula.

**Proposition 3.** *Let $\exp_{Riem} \colon S_n \to \mathrm{SPD}_n$ be the Riemannian exponential map based at $I \in \mathrm{SPD}_n$, and $\exp$ be the matrix exponential. Then $\exp_{Riem} = \exp$.*

*Proof.* Let $U \in S_n$ be a tangent vector to $\mathrm{SPD}_n$ at the basepoint $I$, and orthogonally diagonalize as $U = KDK^T$ for some $K \in O(n)$, $D = \mathrm{diag}(d_1, \ldots, d_n)$. As $D$ is tangent to the maximal flat $\mathcal{D}$ of diagonal matrices, the geodesic segment $\exp_{Riem}(tD)$ must be a geodesic in $\mathcal{D}$, which we know from Lemma 3 to be the coordinate-wise exponential of a straight line in $\mathbb{R}^n$. Precisely, this geodesic is $\mathrm{diag}(\ldots, e^{d_i t}, \ldots)$, and so the original geodesic with initial tangent $U = KDK^T$ is $\exp_{Riem}(tU) = K \, \mathrm{diag}(\ldots, e^{d_i t}, \ldots) K^T$ by Lemma 1. Specializing to $t = 1$, this gives the claim:

$$
\begin{aligned}
\exp_{Riem}(U) &= K \exp_{Riem}(D) K^T \\
&= K \, \mathrm{diag}(\ldots, e^{d_i}, \ldots) K^{-1} \\
&= K \exp(D) K^{-1} \\
&= \exp(KDK^{-1}) \\
&= \exp(U)
\end{aligned}
$$

$\square$

This easily transfers to an understanding of the Riemannian exponential at an arbitrary point $P \in \mathrm{SPD}_n$, if we identify the tangent space at $P$ with the symmetric matrices $S_n$ as well.

**Corollary 3.** *The exponential based at an arbitrary point $P \in \mathrm{SPD}_n$ is given by*

$$\exp_{Riem,P}(U) = \sqrt{P} \exp(\sqrt{P^{-1}} U \sqrt{P^{-1}}) \sqrt{P}$$

*Proof.* Given $P \in \mathrm{SPD}_n$ and tangent vector $U \in T_P \mathrm{SPD}_n$ identified with the set $S_n$ of symmetric matrices, note that $X \mapsto \sqrt{P^{-1}} X \sqrt{P^{-1}}$ is a symmetry of $\mathrm{SPD}_n$ taking $P$ to $I$ and $U$ to $\sqrt{P^{-1}} U \sqrt{P}^{-1}$. Using the fact that we understand the Riemannian exponential at the basepoint, we see $\exp_{Riem}(\sqrt{P^{-1}} U \sqrt{P^{-1}}) = \exp(\sqrt{P^{-1}} U \sqrt{P^{-1}})$. It only remains to translate the result back to $P$, giving the claimed formula. $\square$

**Proposition 4.** *Let $\log_{Riem} \colon \mathrm{SPD}_n \to S_n$ be the Riemannian logarithm map based at $0 \in S_n$, and $\log$ be the matrix logarithm (note that while the matrix logaritm is multivalued in general, it is uniquely defined on $S_n$). Then $\log_{Riem} = \log$.*

*Proof.* Defined as the inverse of $\exp_{Riem}$, the Riemannian logarithm must satisfy

$$\log_{Riem} \circ \exp_{Riem} = \mathrm{id}_{S_n}$$

Let $U \in S_n$ and orthogonally diagonalize as $U = KDK^T$. Applying the Riemannian exponential, we see $\log_{Riem}(K \exp(D) K^T) = KDK^T$. Recalling from Lemma 3 the relation between isometries of $\mathrm{SPD}_n$ and their application on tangent vectors, we see that we may rewrite the left hand side as $\log_{Riem}(K \exp(D) K^T) = K \log_{Riem}(\exp(D)) K^T$. Appropriately cancelling the factors of $K, K^T$ we arrive at the relationship

$$\log_{Riem}(\exp(D)) = D.$$

That is, restricted to the diagonal matrices, the Riemannian logarithm is an inverse of the matrix exponential, so Riemannian log equals matrix log. Re-absorbing the original factors of $K$ shows the same to be true for any positive definite symmetric matrix; thus $\log_{Riem} = \log$. $\square$

As for the exponential, conjugating by a symmetry moving $I$ to an arbitrary point $P$, we may describe the Riemannian logarithm at any point of $\mathrm{SPD}_n$.

**Corollary 4.** *The logarithm based at an arbitrary point $P \in \mathrm{SPD}_n$ is given by*

$$\log_{Riem,P}(Q) = \sqrt{P} \log(\sqrt{P^{-1}} Q \sqrt{P^{-1}}) \sqrt{P}$$

### C.4 Vector-valued Distance

Here we collect useful observations about the vector-valued distance on $\mathrm{SPD}_n$, culminating in a proof of the fact that it is a complete invariant of pairs of points, as claimed in §3.

**Proposition 5.** *The vector-valued distance is well-defined: given any pair $P, Q \in \mathrm{SPD}_n$ of points and any two isometries taking $P, Q$ to the basepoint, a diagonal matrix respectively, the diagonal matrices differ at most by a permutation of their entries.*

*Proof.* We see heuristically that there is no remaining continuous degree of freedom by dimension count: the isometry group $GL(n; \mathbb{R})$ has dimension $n^2$, and we require $\dim(\mathrm{SPD}_n) = n(n+1)/2$ degrees of freedom to translate $P$ to the origin, and a further $\dim O(n) = n(n-1)/2$ degrees of freedom to diagonalize the image of $Q$ while fixing $I$. As $\dim GL(n; \mathbb{R}) = \dim \mathrm{SPD}_n + \dim O(n)$, there are no remaining continuous degrees of freedom. To see that the remaining ambiguity is precisely permutation of coordinates, note that conjugating a diagonal matrix by an orthogonal matrix results in another diagonal matrix only if the conjugating matrix is a permutation matrix. $\square$

**Proposition 6.** *If two points $P, Q \in \mathrm{SPD}_n$ have the same vector-valued distance from the basepoint $I$, then there is an isometry fixing $I$ taking $P$ to $Q$.*

*Proof.* For two matrices to have the same vector-valued distance from $I$ is equivalent to those two matrices having the same set of eigenvalues. Let $\lambda_1, \ldots, \lambda_n$ be a list of these eigenvalues with multiplicity, and construct two orthonormal bases $(v_i), (w_i)$ of $\mathbb{R}^n$ as follows. For each $i$, let $v_i$ be an eigenvector of $P$ with eigenvalue $\lambda_i$, and $w_i$ an eigenvector of $Q$ with eigenvalue $\lambda_i$ (in the case the eigenvalues are distinct, such bases are unique up to flipping the sign of each vector, but nontrivial choices must be made in the case of coincident eigenvalues). Given this pair of orthonormal bases, let $K \in O(n)$ be the orthogonal matrix which takes $(v_i)$ to $(w_i)$. It is then an easy observation of linear algebra to note that $Q = KPK^{-1}$, but recalling $K^T = K^{-1}$ we see this is interpreted in the geometry of $\mathrm{SPD}_n$ to say that there is an isometry $X \mapsto KXK^T$ fixing $I$ and taking $P$ to $Q$. $\qquad\square$

Combining Propositions 5 and 6, after translating appropriately to the basepoint yields the following cornerstone of the theory, showing the vector-valued distance to be the *best possible* invariant.

**Corollary 5.** *The vector-valued distance is a complete invariant of pairs of points. Two pairs of points $(P, Q)$ and $(P', Q')$ cam be mapped to one another by an isometry if and only if they have the same vector-valued distance.*

It's important to note that while the vector-valued distance is not *literally* a metric distance (it is vector valued, instead of positive-real-number valued, for one) it enjoys some properties analogous to traditional metric distances. For a brief review of some of these (the vector-valued triangle inequality, etc) see Kapovich, Leeb & Porti. [46], and Kapovich, Leeb & Millison [45].

One property distinguishing the vector-valued distance from traditional metrics is its assymmetry. We will wish to recall this relationship later on, and so prove it here for completeness.

**Lemma 4.** *For $P, Q \in \mathrm{SPD}_n$, the vector-valued distance satisfies*

$$d_{vv}(P, Q) = -d_{vv}(Q, P)$$

*with equality understood up to permutation of coordinates.*

*Proof.* The computation of $d_{vv}(P, Q)$ differs from that of $d_{vv}(Q, P)$ in the first step, where we reduce it to computing a function of the eigenvalues of $P^{-1}Q$ or $Q^{-1}P$ respectively. Noting these are inverses of one another, their eigenvalues are reciprocals we may perform the following calculation, where $\{\lambda_i(X)\}$ denotes the eigenvalues of $X$.

$$
\begin{aligned}
d_{vv}(Q, P) &= \log(\ldots, \lambda_i(Q^{-1}P), \ldots) \\
&= \log(\ldots, \lambda_i((P^{-1}Q)^{-1}), \ldots) \\
&= \log(\ldots, \lambda_i((P^{-1}Q)^{-1}), \ldots) \\
&= \log\left(\ldots, \frac{1}{\lambda_i((P^{-1}Q))}, \ldots\right) \\
&= -\log(\ldots, \lambda_i((P^{-1}Q)), \ldots) \\
&= -d_{vv}(P, Q)
\end{aligned}
$$

$\qquad\square$

### C.5 Riemannian Distance

This Riemannian metric allows the computation of the length of curves $\gamma \colon [0, 1] \to \mathrm{SPD}_n$ as

$$\mathrm{length}(\gamma) = \int_0^1 \sqrt{\langle \gamma'(t), \gamma'(t) \rangle_{\gamma(t)}} \, dt.$$

This in turn induces a distance function $d : \mathrm{SPD}_n \times \mathrm{SPD}_n \to \mathbb{R}$, by taking the infimum of the lengths of all paths joining two points:

$$d(P, Q) = \inf_{\substack{\gamma \colon [0,1] \to \mathrm{SPD}_n \\ \gamma(0)=P, \, \gamma(1)=Q}} \left\{ \mathrm{length}(\gamma) \right\}$$

While for general Riemannian manifolds such a distance function may be impossible to explicitly compute, the symmetries of $\mathrm{SPD}_n$ provide a readily computable formula.

**Proposition 7.** *The Riemannian distance from the basepoint $I$ to a point $P \in \mathrm{SPD}_n$ is given by $d(I, P) = \sqrt{\sum_{i=0}^n \log(\lambda_i(P))}$ where $\{\lambda_i(P)\}$ are the eigenvalues of of $P$.*

*Proof.* Let $P \in \mathrm{SPD}_n$ be arbitrary, and orthogonally diagonalize as $P = KDK^T$. As $K \in O(n)$, the isometry $X \mapsto KDK^T$ fixes $I$, so the distance $d(I, P)$ equals the distance $d(I, D)$. Note as this action of $K$ is by conjugacy, the diagonal entries $d_i$ of $D$ are precisely the eigenvalues of $P$. As $D$ lies in the totally geodesic Euclidean subspace $\mathcal{D}$, this distance is realized by the unique Euclidean geodesic connecting $I$ to $D$. Using Lemma 2, we may translate to familiar coordinates on $\mathbb{R}^n$ and notice this is the distance from the origin $0$ to the point $x = (\log(d_1), \ldots, \log(d_n))$. That is, $d(I, D) = \sqrt{\sum_i \log(d_i)^2}$ as claimed. $\square$

This immediately generalizes to the distance between a pair of arbitrary points, via conjugating by a symmetry moving one to the origin. However, with a little more work one may get a simpler expression for the general distance.

**Proposition 8.** *The Riemannian distance between two arbitrary points $P, Q \in \mathrm{SPD}_n$ is given by $d(P, Q) = \sqrt{\sum_i \log(\lambda_i(P^{-1}Q))}$ where $\{\lambda_i(P^{-1}Q)\}$ are the eigenvalues of of $P^{-1}Q$.*

*Proof.* If $P, Q$ are arbitrary points in $\mathrm{SPD}_n$, we may use an isometry to translate $P$ to the basepoint, while simultaneously moving $Q$ to $R = \sqrt{P^{-1}}Q\sqrt{P^{-1}}$. As isometries preserve distances, we have $d(P, Q) = d(I, R)$, and by Proposition 7, this distance is completely determined by the eigenvalues of $R$. As these are invariant under conjugacy, we replace $R$ with its conjugate by $\sqrt{P^{-1}}$ to get the matrix

$$
\begin{aligned}
R' &= \sqrt{P}R\sqrt{P}^{-1} \\
&= \sqrt{P^{-1}}\sqrt{P^{-1}}Q\sqrt{P^{-1}}\sqrt{P} \\
&= P^{-1}Q
\end{aligned}
$$

$\square$

### C.6 Finsler Distances

The Riemannian distance function on a manifold is completely determined by its Riemannian metric, a choice of inner product on the tangent bundle. Generalizing this, Finsler metrics are the class of distance functions which may be constructed from a smoothly varying choice of norm $\|\cdot\|_F$ on the tangent bundle (which need not be induced by an inner product). The basic theory proceeds in direct analogy to the Riemannian case: the length of a curve $\gamma\colon [0,1] \to \mathrm{SPD}_n$ with respect to a Finsler metric is still defined via integration of this norm along the path, and the distance between points by the infimum of this over all rectifiable curves joining them

$$
\mathrm{length}_F(\gamma) = \int_0^1 \|\gamma'\|_F dt, \qquad d_F(P, Q) = \inf_{\substack{\gamma\colon [0,1]\to\mathrm{SPD}_n \\ \gamma(0)=P,\, \gamma(1)=Q}} \left\{ \mathrm{length}_F(\gamma) \right\}
$$

The geometry of $\mathrm{SPD}_n$ allows the computaiton of all Finsler metrics directly from the vector-valued distance. As Riemannian metrics are in particular a special case of Finsler metrics, we begin by recasting our previous observations in this light. In §C.5 we derived a formula for the Riemannian distance function directly from the infintesimal Riemannian metric. But in light of Corollary 5, since the Riemannian distance is a function which depends only on its input points up to isometry, it must also be recoverable from the vector-valued distance. Indeed, looking at Proposition 7 we see there is a simple rephrasing to this effect:

**Corollary 6.** *The Riemannian distance from the basepoint $I$ to an arbitrary point $P \in \mathrm{SPD}_n$ is the Euclidean norm of the vector-valued distance from $I$ to $P$.*

One of the great advantages of higher rank symmetric spaces is the generalizations to which this rephrasing lends itself. Namely, the Euclidean metric is not special in this construction, and any sufficiently symmetric norm on $\mathbb{R}^n$ can induce a distance function on $\mathrm{SPD}_n$ in this way.

**Proposition 9.** *Let $\| - \|$ be any norm on $\mathbb{R}^n$ which is invariant under the permutation of coordinates. Then $\| - \|$ induces a distance function $d$ on $\mathrm{SPD}_n$ by*

$$d(P, Q) = \|d_{vv}(P, Q)\|$$

*Proof.* We first note this function is well-defined, as by Proposition 5 the vector-valued distance of $(P, Q)$ is well-defined up to permutation of coordinates, and our norm is invariant under this symmetry by hypothesis. To see that $d$ is in fact a distance function on $\mathrm{SPD}_n$, we now need to show it satisfies the axioms of a metric:

1. $d(P, Q) \geq 0$, $d(P, Q) = 0 \implies P = Q$

2. $d(P, Q) = d(Q, P)$

3. $d(P, R) \leq d(P, Q) + d(Q, R)$

To check the identity of indescernibles (1), note that $d$ is necessarily nonnegative as $\| - \|$ is, and if $d_{(}P, Q) = 0$ then the norm of $d_{vv}(P, Q)$ is zero, so the vector-valued distance itself is zero. But as this is a complete invariant and $d_{vv}(P, P) = 0$, this means $P = Q$.

Note that symmetry (2) is not automatic as the vector-valued distance itself is asymmetric. However recalling Lemma 4, we see that changing the order causes only a global negative sign, and the central symmetry of $\| - \|$, as a virtue of being a norm, gives equality.

The triangle inequality (3) is more subtle, and requires an understanding of the triangle inequality for the vector-valued distance. See the dissertation of Planche [66], Chapter 6 and the work of Kapovich, Leeb and Millson [45] for details. $\qquad\square$

For our experiments, the most important such distances are induced by the $\ell_1$ and $\ell_\infty$ norms on $\mathbb{R}^n$. For completeness, the resulting distance functions are described below.

**Proposition 10.** *The distance function induced from the $\ell_1$ metric applied to the vector-valued distance can be computed as $d_{F_1}(P, Q) = \sum_{i=1}^n |\log \lambda_i(P^{-1}Q)|$, where $\lambda_i(P^{-1}Q)$ runs over the eigenvalues of $P^{-1}Q$.*

*Proof.* The vector-valued distance $d_{vv}(P, Q)$ is the vector of logarithms of the eigenvalues of $R = P^{-1}Q$, and its $\ell^1$ norm is the sum of their absolute values:

$$\|(\log(\lambda_1(R), \ldots, \lambda_n(R))\|_{\ell^1} = \sum_{i=1}^n |\log \lambda_i(R)|$$

where $\lambda_i(R)$ is the $i^{th}$ eigenvalaue of $R$, in decreasing order. $\qquad\square$

A similar calculation yields the formula for the $F^\infty$ distance function.

**Proposition 11.** *The distance function induced from the $\ell_\infty$ metric applied to the vector-valued distance can be computed as $d_{F_\infty}(P, Q) = \lambda_1(P^{-1}Q)$ where $\lambda_1(-)$ returns the largest eigenvalue of the input matrix.*

### C.7 Relations with Other Metrics

Other distances previously used in the literature can be reconstructed from the vector-valued distance, by applying a suitable function:

The *Affine Invariant metric* of [65] is nothing but the usual Riemannian metric discussed in §C.5.

The *symmetric Stein divergence* [71], is given by

$$S(P, Q) := \log \det \frac{P + Q}{2} - \frac{1}{2} \log \det(PQ)$$

This can be computed from the vector-valued distance

$$d_{vv}(P, Q) = \log(\lambda_1(P^{-1}Q), \ldots, \lambda_n(P^{-1}Q))$$

by applying the function

$$\|v\|_S = \sum_{i=1}^n \log \frac{e^{-v_i/2} + e^{v_i/2}}{2}.$$

Indeed

$$
\begin{aligned}
S(P,Q) \quad &= \log\det \tfrac{P+Q}{2} - \tfrac{1}{2}\log\det(PQ) \\
&= \log\det P\left(\tfrac{\mathrm{Id}+P^{-1}Q}{2}\right) - \log\det(P\sqrt{P^{-1}Q}) \\
&= \log\det \tfrac{\mathrm{Id}+P^{-1}Q}{2} - \log\det(\sqrt{P^{-1}Q}) \\
&= \sum_{i=1}^n \log\lambda_i\left(\tfrac{\mathrm{Id}+P^{-1}Q}{2\sqrt{P^{-1}Q}}\right) \\
&= \sum_{i=1}^n \log\left(\tfrac{\lambda_i(P^{-1}Q)^{-1/2}+\lambda_i(P^{-1}Q)^{1/2}}{2}\right)
\end{aligned}
$$

In particular we obtain, thanks to the vector-valued distance, a more direct proof of [72].

Instead the *Log-Euclidean* metric $d_{LE}$ [5, 6] is flat, and as such doesn't reflect the curved geometry of SPD. More precisely $d_{LE}$ is the pushforward, through the exponential map $\exp_{Riem} : S_n \to \mathrm{SPD}$ of the Euclidean metric on $S_n$. As a result, for this choice $(\mathrm{SPD}_n, d_{LE})$ is *isometric* to the flat manifold $S_n$. Since the group $GL(n,\mathbb{R})$ does not act by isometries on $(\mathrm{SPD}_n; d_{LE})$, and the distance is therefore not related to the vector-valued distance nor can be computed from it.

Similarly the *Bures-Wasserstein* metric $d_{BW}$ inspired from quantum information theory [13] leads to a non-negatively curved manifold, and thus, again, has a different isometry group. More precisely

$$d_{BW}(P,Q) = \sqrt{\mathrm{tr}(P) + \mathrm{tr}(Q) - 2\sqrt{\mathrm{tr}(PQ)}}.$$

It is computed in [13, Page 15] that the group of isometries of $(\mathrm{SPD}_n, d_{BW})$ is reduced to $O(n)$. As a result, once again, $d_{BW}$ cannot be reconstructed from $d_{vv}$.

# D   Gyrocalculus

A primary difficulty of building analogs of Euclidean quantities in curved spaces is the lack of a vector space structure, making the translation of operations like vector addition or scalar multiplication difficult to immediately interpret. The need for these is already well-noted stumbling block in hyperbolic geometry, as any algorithm using the Euclidean addition of points cannot be implemented directly (for example considering the Poincare disk model, the sum of two points in the disk need not lie in the disk: and even when it does, the result is rarely geometrically meaningful). To combat this, means of interfacing with hyperbolic geometry using "vector-space-like" operations was developed by Ungar [81], which provides an analog of addition $\oplus \colon \mathbb{H}^n \times \mathbb{H}^n \to \mathbb{H}^n$ and of scalar multiplication $\otimes \colon \mathbb{R} \times \mathbb{H}^n \to \mathbb{H}^n$ called 'gyro-addition' and 'gyro-scalar multiplication' respectively. We give a brief introduction to this general theory below, see Ungar's treatment from the lens of differential geometry [80] for further information.

## D.1   Gyrogroups

Gyrogroups are a generalization of groups which encode algebraically some of the geometric properties of symmetric spaces. More precisely, a gyrogroup structure on a set $G$ is given by a binary operation $\oplus$, which is assumed to have an identity element $0 \in G$ and left inverses $\ominus g$ for each $g \in G$. Keeping with the conventions familiar from arithmetic, we write $a \ominus b$ to mean $a \oplus (\ominus b)$. The crucial difference from group theory is that $\oplus$ is *not* required to be associative. Instead, the additional structure of a *gyration operator* $\mathrm{gyr} \colon G \times G \to \mathrm{Aut}(G)$ captures the nonassociativity of $\oplus$ by

$$a \oplus (b \oplus c) = (a \oplus b) \oplus \mathrm{gyr}(a,b)c$$

.

For $(G, \oplus, \mathrm{gyr})$ to form a gyrogroup, an additional axiom is imposed on this gyration, namely that it satisfy the *left loop identity*, $\mathrm{gyr}(a,b) = \mathrm{gyr}(a \oplus b, b)$.

Gyrogroups generalize groups in the sense that every group $G$ is a gyrogroup with its usual binary operation as $\oplus$, and trivial gyration. As with standard groups, it is helpful to have at one's disposal a collection of elementary deductions from these axioms, which may significantly simplify further calculations.

**Proposition 12.** *The identity of a gyrogroup is unique, every left inverse is also a right inverse, and every element has a unique (left, and hence also right) inverse.*

See [80] §3 for a proof of this proposition, which uses only the axioms of a gyrogroup. It can be shown that when a gyrogroup structure exists on a set $G$, it is determined by the operation $\oplus$ alone, in the sense that for any $a, b, c$ we have

$$\text{gyr}(a, b)c = (\ominus(a \oplus b)) \oplus (a \oplus (b \oplus c)) \tag{11}$$

We record also useful properties of the gyration operator following from this, which simplify calculation.

**Proposition 13.** *The following gyrations are trivial: the gyration of any element with zero* $\text{gyr}(0, a) = \text{gyr}(0, a) = \text{gyr}(\ominus a, a) = \text{id}_G$, *or with its inverse* $\text{gyr}(\ominus a, a) = \text{gyr}(a, \ominus a) = \text{id}_G$. *A useful consequence of these is the* nested gyration identity*:*

$$\text{gyr}(a, \ominus \text{gyr}(a, b)b) \, \text{gyr}(a, b) = \text{id}_G$$

These are also proven in [80] §3 , and follow directly from the axioms of a gyrogroup.

Because of the additional complexity of $\oplus$ compared to the binary operation of a standard group, it is often useful in applications to introduce a second binary operation, the *gyrogroup co-operation* $\boxplus$ and its inverse $\boxminus$, defined by

$$a \boxplus b = a \oplus \text{gyr}(a, \ominus b)b \qquad a \boxminus b = a \boxplus \ominus a$$

This operation provides a useful shorthand for solving equations in gyrogroups, which we discuss in D.2.1.

Finally, we give a means of computing the VVD in terms of these operations as claimed in §4.

**Proposition 14.** *The vector-valued distance from $P$ to $Q$ is the vector of logarithms of the eigenvalues of $(\ominus P) \oplus Q$.*

*Proof.* This is the matrix $(\ominus P) \oplus Q = P^{-1} \oplus Q = \sqrt{P^1}Q\sqrt{P^{-1}}$, which is conjugate to $P^{-1}Q$ (as in 8), and so has the same eigenvalues. But the logarithm of these eigenvalues is precisely the vector value distance as defined in §3.1. $\square$

## D.2 Gyro-vector Spaces

Though the operation $\oplus$ is not commutative in the usual sense, a gyrogroup $G$ is called *gyro-commutative* if it commutes *up to gyrations*: ie for every $a, b \in G$, $a \oplus b = \text{gyr}(a, b)(b \oplus a)$. It is within this restricted class of gyro-commutative gyrogroups that a satisfactory analog of familiar vector space operations can be constructed [82].

A gyrovector space is a gyro-commutative gyrororup $(G, \oplus)$ together with a scalar multiplication $\otimes \colon \mathbb{R} \times G \to G$ such that 1 acts as the identity, and its interaction with standard multiplication, gyro-addition and gyration are constrained by

$$
\begin{aligned}
r_1 \otimes (r_2 \otimes a) &= r_1 r_2 \otimes a \\
(r_1 + r_2) \oplus a &= (r_1 \otimes a) \oplus (r_2 \otimes a) \\
r \otimes \text{gyr}(a, b)c &= \text{gyr}(a, b)(r \otimes c) \\
\text{gyr}(r_1 \otimes a, r_2 \otimes a) &= I
\end{aligned}
\tag{12}
$$

Typically a gyrovector space is also assumed to be constructed within an ambient real inner product space, and there are additional compatibility relations between the operations of $(G, \oplus, \otimes)$ and the ambient vector space addition $(+)$ and norm $\|v\| = \sqrt{v \cdot v}$.

Gyro-vector spaces generalize vector spaces much as gyro-groups generalized groups: every vector space is a gyro-vector space with trivial gyration. As such, the formalism of gyro-vector spaces provides a convenient generalization where one may attempt to replace $+, -, \times$ in formulas familiar from Euclidean spaces with $\oplus, \ominus, \otimes$; being careful to recall that gyro-addition is neither commutative nor associative, and gyro-multiplication rarely distributes over $\oplus$.

### D.2.1 Solving Equations in Gyrogroups

As an example of the difficulties posed by this, if one requires the solution to the Euclidean equation $a + x = b$, it is equally correct to write $x = b - a$ or $x = -a + b$. But the translations $x = b \ominus a$ and $x = \ominus a \oplus b$ into a gyrogroup $G$ need not be equal, and generically only the latter solves the gyrovector equation $a \oplus x = b$.

To make this more systematic, note that working inwards respecting the order of operations, we are able to solve any equation in a gyrogroup if we compute a *left cancellation law*, *right cancellation law* and *invert scalar multiplication*.

**Proposition 15** (Left-Cancellation)**.** *Let $a, b$ be elements of a gyrogroup $G$. Then the relation $a \oplus x = b$ is satisfied by the unique value $x = (\ominus a) \oplus b$.*

*Proof.* Substituting the claimed expression for $x$, we verify by direct computation from the axioms of a gyrogroup, and the basic properties of Propositions 12.

$$
\begin{aligned}
a \oplus x &= a \oplus ((\ominus a) \oplus b) \\
&= (a \oplus \ominus a) \oplus \operatorname{gyr}(a, \ominus a)b \\
&= 0 \oplus \operatorname{gyr}(a, \ominus a)b \\
&= id_G(b) \\
&= b
\end{aligned}
$$

$\square$

**Proposition 16** (Right-Cancellation)**.** *Let $a, b$ be elements of a gyrogroup $G$. Then the relation $x \oplus a = b$ is satisfied by the unique value $x = b \boxminus a = a \ominus \operatorname{gyr}(a, \ominus b)b$, where $\boxminus$ is the additive inverse of the gyrogroup co-operation from Section D.1.*

*Proof.* To begin, we put the proposed solution $b \boxminus a$ in a more usable form:

$$
\begin{aligned}
b \boxminus a &= b \boxplus \ominus a \\
&= b \oplus \operatorname{gyr}(b, \ominus \ominus a) \ominus a \\
&= b \ominus \operatorname{gyr}(b, a)a
\end{aligned}
$$

We now verify the claim by subsituting the given value of $x$, and using the properties described in Propositions 12 and 13, (in particular, in the third step we expand $a$ using nested gyration)

$$
\begin{aligned}
x \oplus a &= (b \boxminus a) \oplus a \\
&= (b \ominus \operatorname{gyr}(b, a)a) \oplus id_G(a) \\
&= (b \ominus \operatorname{gyr}(b, a)a) \oplus (\operatorname{gyr}(b, \ominus \operatorname{gyr}(b, a)a) \operatorname{gyr}(b, a)a) \\
&= b \oplus (\ominus \operatorname{gyr}(b, a)a \oplus \operatorname{gyr}(b, a)a) \\
&= b \oplus 0 \\
&= b
\end{aligned}
$$

$\square$

**Proposition 17** (Inverting Scalar Multiplication)**.** *Let $r \in \mathbb{R}$ be any scalar, and $a$ an element of a gyrogroup $G$. Then the relation $r \otimes x = a$ is satisfied by the unique element $x = \left(\frac{1}{r}\right) \otimes a$.*

*Proof.* Substituting $x$ immediately yeidls the result given the axioms of gyro-scalar multiplication:

$$
\begin{aligned}
r \otimes &= r \otimes \left(\frac{1}{r} \otimes a\right) \\
&= \left(r \times \frac{1}{r}\right) \otimes a \\
&= 1 \otimes a \\
&= a
\end{aligned}
$$

$\square$

These three cancellation laws allow one work correctly with the gyro-translations of Euclidean vector space statements. Take for example the vector space expression $a + rx + b = c$ for vectors $a, b, c, x$ and scalar $r$. One possible gyro-vector space translation of this is $(a \oplus (r \otimes x)) \oplus b = c$ — and given this translation, we may work fully within the gyrovector space to solve for $x$ as follows:

$$(a \oplus (r \otimes x)) \oplus b = c$$
$$a \oplus (r \otimes x) = c \boxminus b$$
$$r \otimes x = (\ominus a) \oplus (c \boxminus b)$$
$$x = \frac{1}{r} \otimes ((\ominus a) \oplus (c \boxminus b))$$