# OpenReview forum: "Vector-valued Distance and Gyrocalculus on the Space of Symmetric Positive Definite Matrices"
_NeurIPS.cc/2021/Conference — NeurIPS 2021 Spotlight_

### Official Review · Reviewer_rEf9 · 2021-07-15

**Rating:** 7
**Confidence:** 3

**Summary:**

The paper firstly defines a vector valued metric for SPD matrices and then it addresses the issue of defining operations for a set of SPD matrices (addition, matrix multiplication, scalar multiplication) using gyrocalculus. This operations allow to make computations on a set of SPD matrices, showing how the setting combined with different methods outperform the state of the art in three case studies.

**Ethical Concerns:**

No Ethical concerns

**Limitations And Societal Impact:**

Th authors listed some limitation of their work but it should be a bit enlarged in the conclusion.

**Main Review:**

Positive aspects: the papar address the urgent and important problem of analyzing non-euclidean objects such as spd matrices in a correct geometrical and algebraic framework. In addition, SPD matrices are broadly used in different applications, so the potential impacts of the work are high. The paper is overall well written.

Negative aspects: If I have correctly understood, I am concerns about the fact that the operations defined (Section 4) relies on a basepoint, thus they are local. This is causing two main problems: (a) how the framework behave if the points are not close to each other, i.e. is the tangent space a good approximation? (b) how the framework behave close to the singularity points of the space? If it is solved by the gyrocalculus approach, it should be better explained.

Major comments:
Section 3. There are several works regarding the comparison between metrices on SPD matrices. Is the vector valued metric intrisic or extrinsic? Please detailed it.

Section 2. How does this work relate with the work on Equivariant neural networks?

Section 4. The space of SPDn have singularities as well as positive curvature. In this area of the space, the approximation of the space with tangent space is not a good choice (e.g. the origin has not one unique tangent space). Could the author clarify how the framework interact with this problem?

Section 4. The gyro-vector space structure is based on the choice of a basepoint. The authors should better state where is the improvement in definition the operation in Section 4 - which relies anyway on the tangent space approach and the base point choice - over the usage of the standard operation on the tangent space (R^(n-1)n/2).


Section 6 As a common behave in the literature, the methodology used are given for granted. It should be useful to understand how the operations well behave with respect to different models used in the section, by enlarging this introduction.

Section 6.1 The example is good but describing a graph in the space of SPDn is a choice which could rise some concerns. In addition, it is not clear to me which is the graph representation. Is the graph represented as a Laplacian? if so then it is an SPD matrix, otherwise the example it is not clear to me.


Minor comments:
Section 3.1 why vector-valued distance is better? it is not clear to all readers and it could sound like an unusual choice.

Section 3.1 why 'the VVD provides much more information than just the distance'?

Section 6.1 Baselines: the results reported for the other methods are taken from the original papers. Is the setting exactly the same? It is often a risky choice in terms of reproducibility.

Tables in Section 6 are hard to read

**Time Spent Reviewing:**

4

---

> ### Author Response · Authors · 2021-08-09
> **Response Reviewer rEf9**
>
> Regarding the choice of a basepoint:
> The space SPD is a symmetric space. This implies in particular that the group invertible matrices act transitively on SPD, so any point can be mapped to any other point in this space. Therefore, the choice of basepoint is irrelevant. Choosing a different basepoint is the same as applying a global isometry. The space has no singularity points. Furthermore, SPD is a Riemannian manifold of nonpositive curvature, therefore the exponential map gives a homeomorphism between the tangent space at any chosen basepoint and SPD. Hence, we can use calculations in the tangent space without causing any distortions of the geometry.
>
> Regarding the positive curvature of SPD and singularities in the space:
> SPD is a Riemannian manifold of nonpositive curvature, it has no regions of positive curvature. It also has no singularities. In fact, the neighborhood of any point looks the same in this space.
> The extrinsic view of SPD as a cone might be misleading: every interior point of this cone is a point in SPD, and in fact, the neighborhood of any point looks the same. The tip of the cone, the zero matrix, or any semi-positive definite matrix of lower rank in the boundary of the cone is not a point in SPD.
>
> Regarding the advantages of the vector-valued distance function and the information it provides, please refer to the general response.
> In concrete, the vector-valued distance provides an **intrinsic metric**.
>
> Equivariant neural networks are applied in problems where, if we transform the input (translation, rotation, etc), then also the output is transformed in the same way. An example is the problem of semantic segmentation in computer vision. In this case, if the input image is transformed (rotated, for example), then the output pixel-wise segmentation mask must transform in the same way as the input image. Equivariant neural networks devise ways to do parameter sharing and exploit groups of symmetries for this type of task (see Gerken et al, “Geometric Deep Learning and Equivariant Neural Networks”).
> In our case, we develop network layers that are able to scale, translate, rotate or reflect representations on the SPD manifold respecting its geometry. We apply different transformations in the symmetry group of SPD with the goal of learning effective representations for the task at hand (e.g. knowledge base completion, item recommendation, or QA in our paper). However, in this work, we do not pursue the same design goals or applications as equivariant neural networks.
>
> Regarding the knowledge graph representations, we do not represent the graphs as a Laplacian. We follow the approach known as “Knowledge graph embedding method” (according to [30]), in which we learn representations of entities (nodes) as embeddings in SPD, i.e. each entity has an associated SPD matrix embedding in the space (L220). For each relation, we learn two matrices (L228): one is represented as $M_r$ which is the transformation (scaling, rotation or reflection), and the other one (namely $R$) is an embedding in SPD that we add to the result of the transformation. The equivalent approach of learning a representation for each entity, and relation-specific transformations and/or additions is followed by [6, 7, 13, 21, 24, 42, 64, 68, 83], among many others (see Ji et al. “A Survey on Knowledge Graphs: Representation, Acquisition and Applications”).
> For the advantages of choosing SPD as target space for learning graph embeddings, see the general response.
>
> We appreciate the observation about the reproducibility issues that may arise with previous work. However, for the Knowledge graph experiments (S6.1) we follow the common practice of taking the results from the original papers. We replicate the exact same setup described in [21, 42], and compare to the same baselines with the same results of [21].
> For the item recommendation and QA sections, we implement the baselines and run experiments to provide a valid comparison under the exact same setup, conditions, and data for all analyzed models.

---

> > ### Comment · Reviewer_rEf9 · 2021-08-23
> > **Response acknowledged**
> >
> > I thank the authors for the details responses which cleared my doubts about the paper. I keep my positive score regarding the acceptance of the paper.

---

### Official Review · Reviewer_5T8d · 2021-07-24

**Rating:** 7
**Confidence:** 2

**Summary:**

This paper presents a generalized representation on the SPD manifold. The author leverages the vector-valued distance to help obtain unified metrics form on the manifold of SPD and visualize the high-dimensional data. In addition, the author also proposes to equip the SPD space with the arithmetic operations under the gyrovector space structure, which could help implement the calculations directly on the SPD manifold and maintain the essential structural information. Experiments are conducted on three different downstream tasks and demonstrate the effectiveness of the proposed method.

**Limitations And Societal Impact:**

No potential negative societal impact of this work.

**Main Review:**

Strengths:

1. The paper is mostly well presented.
2. Concise description of the proposed method and detailed theoretical analysis are both provided.
3. The experiments cover different kinds of tasks and data types.

Weaknesses:

1. In Sec. 1, the author mentions that "the representational power of SPD is not fully exploited...". It is desirable to provide some details or examples to illustrate why/how the representational power of SPD is under exploited.

2. In Sec. 3.1, on line 101, it is claimed that "In Euclidean or hyperbolic spaces, the relative position between two points is completely  determined by their distance, ...", it is ideal to explain this more clearly.

3. In the experiment section, the under-performance of the proposed method under the rotation and reflection operations are quite significant and thus some insightful explanation will be much helpful.

5. Some typos. On line 215, "the task of is to predict..."




**Time Spent Reviewing:**

6

---

> ### Author Response · Authors · 2021-08-09
> **Response Reviewer 5T8d**
>
> We answer to the weakness found by the reviewer:
>
> 1: Regarding the representational power of SPD being under-exploited:
>
>    * We argue that the full representational power of SPD in many works is not fully exploited since they apply approximation methods that locally flatten the manifold by projecting it to its tangent space [18, 74]. Another line of work [31, 81] embeds SPD points into a higher dimensional reproducing kernel Hilbert space via kernel functions. Moreover, this method ends up converting the SPD matrix into a vector [25]. Both approaches do this with the aim of employing conventional Euclidean classifiers on the data points. However, the projections distort the geometrical structure of the manifold, often resulting in undesirable effects [35], such as the swelling of diffusion tensors [56].
>
>    * Previous work only used a distance function (Riemannian or Finsler distance) on SPD, but we argue that these distances do not represent the full geometry of SPD. The full geometry is however accounted for by taking the vector valued distance (VVD) function. To give an example, geodesics between points in SPD have different regularity, which is related with the number of maximal Euclidean subspaces that contain the geodesic. The Riemannian or Finsler distance functions cannot distinguish the differences between these geodesics of different regularity, but the VVD function can. The unique geodesic between two points is regular if and only if the entries of the VVD vector are pairwise different. Considering the VVD, we thus capture the full geometry, and hence the full representation power of SPD.
>
> 2: For a more detailed explanation of what we mean by pointing out the relative position between two points in different spaces, please refer to the advantages of the VVD in the general response.
>
> 3: We observe the under-performance of rotations and reflections only for the knowledge graph experiments, and not in all cases (for example, SPD reflections outperform the baselines in WN18RR, both with the Riemannian and Finsler one metric). Given that this behavior does not repeat for item recommendations nor QA experiments, we consider this is due to overfitting in some particular cases. Although we tried different regularization methods, we think the main reason for the under-performance is finding a sub-optimal configuration rather than a geometric reason. We appreciate the finding and we will add this observation in the analysis of the results.

---

> > ### Comment · Reviewer_5T8d · 2021-09-02
> > **Response acknowledged**
> >
> > Thanks for the authors' explanations. I keep the positive rating.

---

### Official Review · Reviewer_ErZJ · 2021-07-24

**Rating:** 6
**Confidence:** 3

**Summary:**

In this paper, the authors studied representation learning on SPD manifold by proposing a vector-valued distance function that helps to visualize the embeddings and also generalizing the arithmetic operations to SPD manifold via the Gyrocalculus. They showed good representation power in knowledge graph completion, item recommendation, and question answering.

**Main Review:**

Overall, the paper is well-written with nice visualizations. With the increasing attention on geometric deep learning models, the proposed arithmetic operations on SPD matrices can be a good contribution.

However, the major concern I have is on the experiments. In all three experiments, the authors mainly consider representation learning of different objects on SPD manifold, including node entities in knowledge graph and word embeddings. These objects are primarily modelled on hyperbolic space in the literature with a clear justification, i.e., hierarchical structures that are known to exist in graph entities and between words are captured. However, it is unclear why embedding these objects onto SPD manifold is beneficial.

In addition, as mentioned by the authors, the computational cost can be high to perform the arithmetic operations compared to the hyperbolic counterparts. It would be helpful to report and compare the runtime in the experiments. Because in some applications, the improvements over the baselines are marginal. If the time required to train the models on SPD manifold take much longer, embedding onto SPD manifold is not well-justified.

Instead, it would be better if the authors could compare with other SPD representation learning methods. Also, it would be interesting to see how to construct a neural network architecture on SPD manifold using the proposed arithmetic operations.


**Time Spent Reviewing:**

5

---

> ### Author Response · Authors · 2021-08-09
> **Response Reviewer ErZJ**
>
> From the summary of this review, we would like to recall that the VVD is not only for visualization purposes. It also allows us to implement universal models by reconstructing the Riemannian, or any Finsler distance. Moreover, it provides additional information as explained in the general response.
>
> Regarding the intuition in the choice of SPD and its advantages, SPD captures hierarchical arrangements through hyperbolic subspaces and also “flat” arrangements through Euclidean subspaces. Please refer to the general response for a more detailed explanation.
>
> We observe as the main drawback the computational complexity of the operations (L368). For points of $m$ dimensions in hyperbolic space, the cost of the distance calculation is $\mathcal{O}(m)$. For SPD spaces of rank $n$, the number of **free parameters** is $n(n + 1) / 2$ (L69). The cost of the distance computation is $\mathcal{O}(n^3)$ (Appendix, L696). Note that for $n = 24$, which is a low value of $n$, the SPD space compares to a model of $300$ dims, thus the method can be used in high dimensions.
>
> For the item recommendation experiments, hyperbolic models converged in ~2 hs while SPD models took ~12 hs. For QA experiments, hyperbolic models took ~12 minutes, and SPD took ~90 minutes. We observe that while SPD training is significantly slower compared with vector representations, it is still manageable without requiring an excessive amount of time nor computational resources (all experimental details, setup, and hardware can be found in Appendix B). Moreover, thanks to the fact that we use tangent space optimization, we can employ off-the-shelf optimizers, which allow us to parallelize the training on multiple GPU/CPU environments, in cases of large datasets.
>
> With respect to comparing our work with other SPD representation learning methods: to replicate experiments such as the one we performed with knowledge graphs and item recommendation, we require an arithmetic operation in the space, such as addition, which is not defined in previous work on SPD. Thus, a vis-a-vis comparison of an equivalent model would not be possible. Moreover, the definition of the transformation layers employed in [25, 29, 35] requires optimizing over compact Stiefel manifolds, plus the derivation of a particular Riemannian matrix backpropagation rule, which is a highly non-trivial and impractical approach, whereas our implementation employs off-the-shelf optimizers. Given these reasons, we choose baseline models that replicate the exact same operations (i.e. scaling, rotations, and reflections, in their respective spaces), and under equivalent conditions and settings for optimization.
> Regarding the different proposed metrics, in Appendix C.7 we detail how the VVD generalizes other SPD metrics.
>
> About the comment on constructing neural network architectures on SPD manifolds, that is what the experiments are intended for (L205-208). We explain how the different linear mappings, along with the gyro-vector operations, can be seen as feature transformations and employed as building blocks for SPD neural models (L167-170). The definition of the gyro-addition (S4, L145) allows us to define the equivalent of bias addition. Finally, although we do not employ non-linearities, our approach can be seamlessly integrated with the ReEig layer (adaptation of a ReLU layer for SPD) proposed in [35].

---

> > ### Comment · Reviewer_ErZJ · 2021-09-01
> > **Response acknowledged**
> >
> > I appreciate the detailed responses from the authors. Major concerns have been addressed.

---

### Official Review · Reviewer_PQuu · 2021-08-02

**Rating:** 8
**Confidence:** 4

**Summary:**

The authors propose the use of a so-called vector-valued distance to extract geometric information from the space of symmetric positive definite matrices (SPD), which is a manifold that is ubiquitous in a wide range of applications, and present the accompanying gyrovector calculus which constructs analogs of vector space operations in this curved space. These operations are then applied to a number of tasks including knowledge graph completion, item recommendation, and question answering, with favorable results compared to the corresponding Euclidean and hyperbolic spaces.

**Limitations And Societal Impact:**

Yes, the authors have adequately addressed the limitations and potential negative societal impact of their work.

**Main Review:**

SPD manifolds have been used in a very wide range of applications in computer vision, medical imaging, and machine learning and the study of their geometries and research into their applications is highly active. The work under review fits naturally within this context and should be of interest to a large community of researchers. The paper is very well-written with a clear and thoughtful structure that includes high quality and enlightening illustrations as well as a clear mathematical exposition. The background literature is also quite thorough and comprehensive with the exception of a few very important omissions which I have noted below. The main contribution of the paper is in the examples considered and the experiments that demonstrate the versatility of the approach and the superior expressivity of SPD manifolds equipped with geometries based on the vector-valued distances compared to Euclidean or hyperbolic baselines. As a result, I am pleased to support acceptance of the paper subject to revision based on the comments and suggestions provided below, some of which I consider to be necessary.

Major Comments:

1. On first reading I understood that the authors had introduced the gyro calculus based on the vector-valued distance as a novel mathematical contribution inspired by the hyperbolic analog. After searching the literature, it became clear that the same gyro vector calculus presented here has already been introduced and discussed in the pure mathematics literature. This needs to be made more clear in the text and relevant papers need to be cited and given due credit. Examples include references [1-3] below. Others may be appropriate as well.

[1] T Abe, O Hatori, Generalized gyrovector spaces and a Mazur-Ulam theorem, published in Publ. Math. Debrecen, 87 (2015), 393--413.

[2] O Hatori, Examples and Applications of Generalized Gyrovector Spaces, Results in Mathematics, 71 (2017), 295-317.

[3] S Kim, Gyrovector Spaces on the Open Convex Cone of Positive Definite Matrices, Mathematics Interdisciplinary Research, (2016).

2. Similar to the above comment, the particular vector-valued distance (equation (1)) has been noted in several works as being a fundamental geometric object for a pair of SPD matrices, which very naturally generates a family of Finsler structures that include the Fisher-Rao (affine-invariant) Riemannian metric and Thompson’s part metric on the positive definite cone as special cases. Relevant papers include the standard reference [4] as well as more recent work such as [5], which studies Thompson geometry in this framework.

[4] R Bhatia, On the exponential metric increasing property, Linear Algebra and its Applications, Volume 375, (2003).

[5] C Mostajeran, C Grussler, R Sepulchre, Geometric Matrix Midranges, SIAM Journal on Matrix Analysis and Applications, (2020).

3. Could the authors elaborate on the issues they faced in using Riemannian optimization, which led them to use tangent space optimization instead? Is an explanation available for these observations?

4. Standard references for Riemannian optimization on manifolds include [6] and [7], which are worth citing on line 201.

[6] P.-A Absil, R Mahony, R Sepulchre, Optimization Algorithms on Matrix Manifolds (2008).

[7] N Boumal, B Mishra, P.-A Absil, R Sepulchre, Manopt, a Matlab Toolbox for Optimization on Manifolds, Journal of Machine Learning Research, (2014).

5. The observed superiority of SPD models compared to hyperbolic models in the experiments is interesting and deserves a deeper discussion in my view. Are particular aspects of the geometry relevant for each of the specific examples/tasks considered, or is the geometry useful in the same way in each case?

6. Perhaps relevant to point 5 above, the authors note on lines 75-77 the contents of SPDn including the number of Euclidean subspaces, hyperbolic subspaces, and hyperbolic planes. Please include a citation for this statement.

7. The authors briefly note the computational limitations of their approach towards the end of the paper. I think this should be expanded and discussed in more detail. Particularly, the distance and gyro calculus operations represent quite a significant computational cost in large dimensions due to eigenvalue computations and matrix operations, which may be missed by a quick glance from a casual reader who may get the wrong idea from the word ‘vector’ in the title!

Minor comments:

1. On line 53, the authors state that the Log-Euclidean geometry (Euclidean-based tools in tangent space) ‘distorts the manifold structure in regions far from the tangent space affecting the performance.’ While I understand what the authors intend to communicate here, it is not the ‘manifold geometry’ that is distorted, since the Log-Euclidean geometry is perfectly valid as a metric structure (Riemannian indeed) on this manifold. It just happens not to be the ‘right geometry’ in many cases, so it is the metric structure that is distorted for many practical applications and not the manifold nature of the space.

2. Remove ‘of’ on line 215.

**Time Spent Reviewing:**

7 hours

---

> ### Author Response · Authors · 2021-08-09
> **Response Reviewer PQuu**
>
> 1 & 2: We appreciate the list of omitted works provided by the reviewer, which we will properly include and give due credit to. For gyrocalculus in particular, we cite the novel work of Ungar [71, 72, 73], and its applications in deep learning by Ganea et al [28]. We will include the additional references for gyrovector spaces and Finsler distances.
>
>
> 3: To perform Riemannian optimization, we need tools that allow us to transform the Euclidean gradient (computed through automatic differentiation) into the right geometry. Available implementations only adapt SGD to this setup (see Bécigneul et al [9], Bonnabel [12]), since the one for Riemannian Adam has bugs: https://github.com/geoopt/geoopt/issues/164). We found Riemannian SGD to be less numerically stable: after a few epochs, matrices are not exactly symmetric anymore which makes many methods that work under the symmetry assumption fail. Moreover, this implementation is much slower than the native pytorch optimizers. Therefore, we opted for tangent space optimization, as applying the exponential map to the points to operate in the space was less computationally expensive. As explained in Appendix C.3, this approach does not incur losses in representational power.
>
>
> 5: For the relevant aspects of the SPD geometry, see the general response.
>
>
> 6: Lines 75-77: We should cite the book of Helgason [34].
>
>
> 7: We refer to the computational cost of the method as the main drawback and place it in the conclusions since we consider it is easier to be noted there. In Appendix A.1 we detail the cost of each matrix computation individually, and of the distance and gyrocalculus operations. We propose to move the cost of the distance computation and its related discussion into the main body of the paper, to raise awareness of the computational complexity of the method.
>
>
> Minor comment 1: Yes, it should not say "manifold structure", but "metric structure".

---

> > ### Comment · Reviewer_PQuu · 2021-08-18
> > **Response acknowledged**
> >
> > I am pleased with the response and reiterate my recommendation to accept the paper.

---

### Author Response · Authors · 2021-08-09
**General Response**

We thank the reviewers for their insightful comments! We are glad that the reviewers agree on how the proposed tools can have an impact on a wide range of applications, on the clear exposition of the paper, and on the diversity of tasks and data of the experiments.

In this section we would like to address two issues that repeat across reviewers:

### Advantages of learning embeddings in SPD
Embeddings into SPD manifolds are beneficial because these manifolds have a richer geometry than hyperbolic spaces. **They contain both, Euclidean subspaces and hyperbolic subspaces.** Therefore, they can accommodate hierarchical structure in data sets in the hyperbolic subspaces, while at the same time represent Euclidean features (e.g. grid-like subgraphs and/or structures with many connections). This makes them more versatile than using only hyperbolic or Euclidean spaces, and in fact their different submanifold geometries might be used to identify and disentangle such substructures in graphs.
We will add a deeper discussion of these advantages when we introduce the SPD space in Section 3.

### Advantages of the vector-valued distance function (VVD)
In Euclidean or hyperbolic space, given two points $A, B$ at distance $k$ from each other, and two other points $C, D$ also at distance $k$ from each other, there exists an isometry of the space, i.e. a transformation preserving all geometric quantities, that will map $A$ to $C$ and $B$ to $D$.
In SPD this is not true, since there exists pairs of points $A, B$ at distance $k$ from each other, and $C, D$ also at distance $k$ from each other, such that there is no isometry mapping $A$ to $C$ and $B$ to $D$. In SPD, only if the VVD between two points $A$ and $B$ is the vector $v$, and the VVD between $C$ and $D$ is also $v$, then there exists an isometry mapping $A$ to $C$ and $B$ to $D$.
This means that the relative position between two points in Euclidean or hyperbolic space is completely determined by a number, namely their distance, but the relative position between two points in SPD is determined by a vector, precisely by the vector valued distance.
In applications this means, that given a representation of a graph in SPD we get finer invariants for the relative position between nodes of the graph, which can be useful to analyze the structure of the graph or perform tasks, e.g. for knowledge graph embeddings.

The VVD thus contains much more information than just the distance:

 - Out of the VVD between two points, one can immediately read the regularity of the unique geodesics joining these two points. This is not possible knowing just the Riemannian distance or a Finsler distance.
 - The VVD contains the full information of the Riemannian distance and of all invariant Finsler distances.
 - In this work we use the VVD function as a tool to analyze the learning progress, as well as a visualization tool.

In future work we hope to use it in further ways:

 - Imposing regularity of the geodesics could be a useful constraint in learning the embeddings
 - Use the regularity of "edges/relations" for knowledge graph embeddings as a further similarity proxy to improve the prediction results.

---

### Decision · Program_Chairs · 2021-09-27

**Decision:**

Accept (Spotlight)

**Comment:**

The paper proposes to use vector-valued distances (and calculus) on the manifold of symmetric positive definite matrices to use in machine learning applications.  As all reviewers agree, the paper contains novel insights and interesting applications (with good analysis). While some theoretical insights are well-known in some previous works, the paper does a great job of putting things cleanly for the NeurIPS community.